# Excitotoxic inactivation of constitutive oxidative stress detoxification pathway in neurons can be rescued by PKD1

Julia Pose-Utrilla[1,2], Lucía García-Guerra[1,2], Ana Del Puerto[1,2], Abraham Martín[3], Jerónimo Jurado-Arjona[2,4,12], Noelia S. De León-Reyes[1,13], Andrea Gamir-Morralla[1,2,12], Álvaro Sebastián-Serrano[1,2], Mónica García-Gallo[5], Leonor Kremer[5], Jens Fielitz[6,7], Christofer Ireson[8,14], Mª José Pérez-Álvarez[2,4,9], Isidro Ferrer[2,10], Félix Hernández[2,4], Jesús Ávila[2,4], Marina Lasa[1], Miguel R. Campanero[1,11] & Teresa Iglesias [1,2]

Excitotoxicity, a critical process in neurodegeneration, induces oxidative stress and neuronal death through mechanisms largely unknown. Since oxidative stress activates protein kinase D1 (PKD1) in tumor cells, we investigated the effect of excitotoxicity on neuronal PKD1 activity. Unexpectedly, we find that excitotoxicity provokes an early inactivation of PKD1 through a dephosphorylation-dependent mechanism mediated by protein phosphatase-1 (PP1) and dual specificity phosphatase-1 (DUSP1). This step turns off the IKK/NF-κB/SOD2 antioxidant pathway. Neuronal PKD1 inactivation by pharmacological inhibition or lentiviral silencing in vitro, or by genetic inactivation in neurons in vivo, strongly enhances excitotoxic neuronal death. In contrast, expression of an active dephosphorylation-resistant PKD1 mutant potentiates the IKK/NF-κB/SOD2 oxidative stress detoxification pathway and confers neuroprotection from in vitro and in vivo excitotoxicity. Our results indicate that PKD1 inactivation underlies excitotoxicity-induced neuronal death and suggest that PKD1 inactivation may be critical for the accumulation of oxidation-induced neuronal damage during aging and in neurodegenerative disorders.

---

[1] Instituto de Investigaciones Biomédicas "Alberto Sols", Consejo Superior de Investigaciones Científicas-Universidad Autónoma de Madrid (CSIC-UAM), C/ Arturo Duperier 4, 28029 Madrid, Spain. [2] CIBERNED, Centro de Investigación Biomédica en Red sobre Enfermedades Neurodegenerativas, Instituto de Salud Carlos III, C/ Valderrebollo, 5, 28031 Madrid, Spain. [3] Experimental Molecular Imaging (Molecular Imaging Unit), CIC biomaGUNE, Paseo Miramon, 182, 20009 San Sebastian, Spain. [4] Centro de Biología Molecular "Severo Ochoa" (CSIC-UAM), C/ Nicolas Cabrera 1, 28049 Madrid, Spain. [5] Protein Tools Unit, Centro Nacional de Biotecnologia, Consejo Superior de Investigaciones Científicas (CSIC), C/ Darwin 3, 28049 Madrid, Spain. [6] Experimental and Clinical Research Center (ECRC), Charité-Universitätsmedizin, Max-Delbrück-Center (MDC) for Molecular Medicine in the Helmholtz Association, Berlin, 13125, Germany. [7] Department of Cardiology, Heart Center Brandenburg and Medical University Brandenburg (MHB), Bernau, 16321, Germany. [8] Cancer Research Technology, London, EC1V 4AD, UK. [9] Departamento de Biología (Unidad Docente Fisiología Animal), UAM, C/ Darwin 2, 28049 Madrid, Spain. [10] Instituto de Neuropatología, Hospital Universitario de Bellvitge, C/ Feixa LLarga s/n, 08907 Barcelona, Hospitalet de Llobregat, Spain. [11] CIBERCV, Centro de Investigación Biomédica en Red de Enfermedades Cardiovasculares, Instituto de Salud Carlos III, Madrid, 28029, Spain. [12] Present address: Institute of Physiological Chemistry, University Medical Center, Johannes Gutenberg University Mainz, Hanns-Dieter-Hüsch-Weg 19, 55128 Mainz, Germany. [13] Present address: Centro Nacional de Biotecnología (CSIC), C/ Darwin 3, 28049 Madrid, Spain. [14] Present address: Pharmidex Pharmaceutical Services, 14 Hanover Street, London, W1S 1YH, UK. Julia Pose-Utrilla and Lucía García-Guerra contributed equally to this work. Correspondence and requests for materials should be addressed to T.I. (email: tiglesias@iib.uam.es)

Neuronal death by excitotoxicity is a critical process in numerous human neuropathologies, such as stroke, traumatic brain injury, epilepsy, Alzheimer's disease, Parkinson's disease, Huntington's disease, amyotrophic lateral sclerosis, and multiple sclerosis[1]. Therefore, intervening the mechanistic steps that lead to excitotoxicity may protect the brain in a broad range of acute and chronic central nervous system pathologies.

Excitotoxicity originates by massive release of the excitatory neurotransmitter glutamate. Overstimulation of postsynaptic glutamate receptors, including the ionotropic N-methyl-D-aspartate (NMDA), α-amino-3-hydroxy-5-methyl-4-isoxazole propionic acid (AMPA), and kainic acid (KA) receptors, overexcites neurons and triggers pro-death cascades[1–3]. Primarily, excessive influx of calcium ions occurs, followed by endoplasmic reticulum stress, mitochondrial dysfunction, generation of high levels of reactive oxygen species (ROS), and oxidative stress damage, leading to neuronal death[4–6].

Protein kinase D1 (PKD1), together with PKD2 and PKD3, constitute a family classified within the calcium/calmodulin-dependent protein kinase superfamily[7]. Numerous stimuli activate PKD through well-established pathways. In many cases this activation is transient, but the mechanisms stopping sustained stimulation remain unexplored[7]. Oxidative stress is an important activator of PKD1 in cellular models, but its capacity to activate this kinase in vivo is largely unknown.

Excitotoxic production of ROS elevates death-associated protein kinase (DAPK) activity, which provokes neuronal apoptosis in cerebral ischemia and seizure models[8]. Accordingly, DAPK absence protects neurons from excitotoxic insults[9, 10]. Of note, DAPK can activate PKD in HeLa cells under oxidative stress conditions[11], thus suggesting that PKD activation may contribute to cellular death.

In addition, oxidative stress can also elicit PKD1 activation, determined by Ser916 autophosphorylation, through initial phosphorylation by Abl and Src kinases (at Tyr469 and Tyr93, respectively) and subsequent phosphorylation by protein kinase C delta (PKCδ) of Ser744 and Ser748 (reviewed in ref. [12]; see Scheme in Fig. 1a). This cascade promotes cellular survival in non-neuronal tumor cells, through activation of I kappaB kinase (IKK) and nuclear factor-kappaB (NF-κB) that induces SOD2 transcription, a gene encoding the mitochondrial manganese-dependent superoxide dismutase (MnSOD) involved in ROS detoxification[13–17]. However, the contribution of NF-κB to neuronal physiopathology is highly controversial, being associated to both neuroprotection and neurotoxicity[18]. NF-κB can regulate genes involved either in neuronal survival or in death[19] and there is also some evidence of NF-κB activation by ROS and excitotoxicity in cultured primary neurons[20–22].

To date, to our knowledge there are no studies investigating PKD1 activation by oxidative stress in neurodegeneration animal models or in samples from human disease.

Whether excitotoxic oxidative stress produces PKD1 activation in neurons, and whether this step leads to changes in neuronal NF-κB activity is an important question that remains unanswered. Moreover, the molecular mechanisms involved in PKD inactivation also remain unknown and the contribution of this inactivation to pathophysiological processes has not been investigated. Here we show the existence of a constitutive neuronal PKD1/IKK/NF-κB/SOD2 oxidative stress detoxification pathway that is inactivated by phosphatase-dependent mechanisms during excitotoxic neurodegeneration. Our study demonstrates that PKD1 potentiates neuronal survival by helping neurons to fight against oxidative stress through IKK and NF-κB.

## Results

**Excitotoxicity regulates neuronal PKD activity.** Excitotoxic concentrations of the NMDA receptor (NMDAR) agonist NMDA together with its co-agonist glycine induce neuronal death[23–25]. To investigate whether PKD is activated by excitotoxicity, we stimulated cultured primary mature cortical neurons with NMDA (50 μM) and glycine (10 μM), a treatment referred here as "NMDA", for different time periods and assessed Ser916 autophosphorylation by immunoblot[26] (Fig. 1a, b). PKD basal activity increased 5 min after NMDA addition (Fig. 1b). Strikingly, 30 min and 1 h of treatment decreased p-Ser916 signal markedly below that in control cells (Fig. 1b), indicating a rapid inactivation of PKD. Note that p-Ser916 band appeared as a doublet in unstimulated neurons and that NMDA modified the intensity of both bands (Fig. 1b). Lentiviral transduction of PKD1 or PKD2-specific short hairpin RNA (shRNAs) indicated that the upper and lower bands corresponded to PKD1 and PKD2, respectively, and that the PKD antibody detected mainly PKD1 (Supplementary Fig. 1a). In addition, studies by RT-qPCR showed that PKD1 transcripts were more abundant than those for PKD2 and PKD3 in mature cultured neurons, and that excitotoxicity did not affect their levels (Supplementary Fig. 1b, c) or those of total PKD protein (Fig. 1b), suggesting that the observed results may reflect changes in kinases and phosphatases (PPs) activities rather than PKD degradation.

Importantly, non-excitotoxic doses of NMDA (≤10 μM) failed to alter PKD activity (Supplementary Fig. 1d). Excitotoxicity was confirmed by the processing of full-length (FL) brain Spectrin to breakdown products (BDPs) by calpain, a protease activated through NMDARs overstimulation[27] (Fig. 1b), as well as measuring neuronal viability (Supplementary Fig. 1e). This neuronal death was not blocked by the caspase inhibitor zVAD, suggesting its non-apoptotic nature (Supplementary Fig. 1f). In addition, we detected an increase in ROS production after 1 h of NMDA addition, when PKD appears inactive (Supplementary Fig. 1g), indicating that excitotoxicity-induced ROS production is not paralleled by sustained activation of PKD.

Excitotoxic neuronal death is coupled to stimulation of extrasynaptic NMDARs[28] where GluN2B is the major subunit[29]. To investigate whether GluN2B is required for NMDA-mediated regulation of PKD activity, we used a selective antagonist for this subunit, ifenprodil (IFN)[30]. This inhibitor prevented activation (Fig. 1c) and inactivation (Fig. 1d) of PKD. The generic NMDAR antagonist DL-AP5 also blocked the regulation of PKD activity by NMDA (Fig. 1c, d). Notably, DL-AP5 but not IFN increased PKD basal activity (Fig. 1c, d), suggesting that activity of synaptic NMDARs, likely through GluN2A, may contribute to PKD dephosphorylation. As expected, both antagonists hampered the excitotoxic process, as shown by the absence of Spectrin BDPs after 1 h of NMDA (Fig. 1d).

Overactivation of NMDARs triggers a massive $Ca^{2+}$ influx. Buffering of extracellular $Ca^{2+}$ with EGTA or treating neurons with the $Ca^{2+}$ ionophore A23187 demonstrated that the entry of this ion was evoking PKD activity changes (Supplementary Fig. 1h, i).

In conclusion, excitotoxicity drives a short activation followed by a sharp inactivation of PKD that depends on overstimulation of NMDARs containing GluN2B subunits through $Ca^{2+}$ entry.

**Phosphorylation-dependent excitotoxic regulation of PKD.** To determine the pathway of PKD activation, we analyzed PKC-dependent activatory transphosphorylation of Ser744/Ser748, and found it changed significantly showing a time-course of phosphorylation/dephosphorylation similar to that showed by p-Ser916 (Fig. 1a, b; Supplementary Fig. 1j). In contrast to the

transient PKD activation, excitotoxicity produced a rapid and sustained activation of DAPK along NMDA treatment, as shown by its gradual activatory dephosphorylation on Ser308[9, 10] (Fig. 1b). Together, these results suggest that PKC, rather than DAPK, is the kinase governing PKD activation changes in excitotoxicity.

Under oxidative stress conditions, PKD stimulation mediated by PKCδ phosphorylation is facilitated through Src family tyrosine kinases (reviewed in ref. [12]), where Tyr469 and Tyr93 are phosphorylated by Abl and Src, respectively. To study PKD Src-dependent tyrosine phosphorylation, we generated a novel phosphospecific monoclonal antibody (mAb) targeting phosphorylated Tyr93 (see details in Methods and Supplementary Fig. 2).

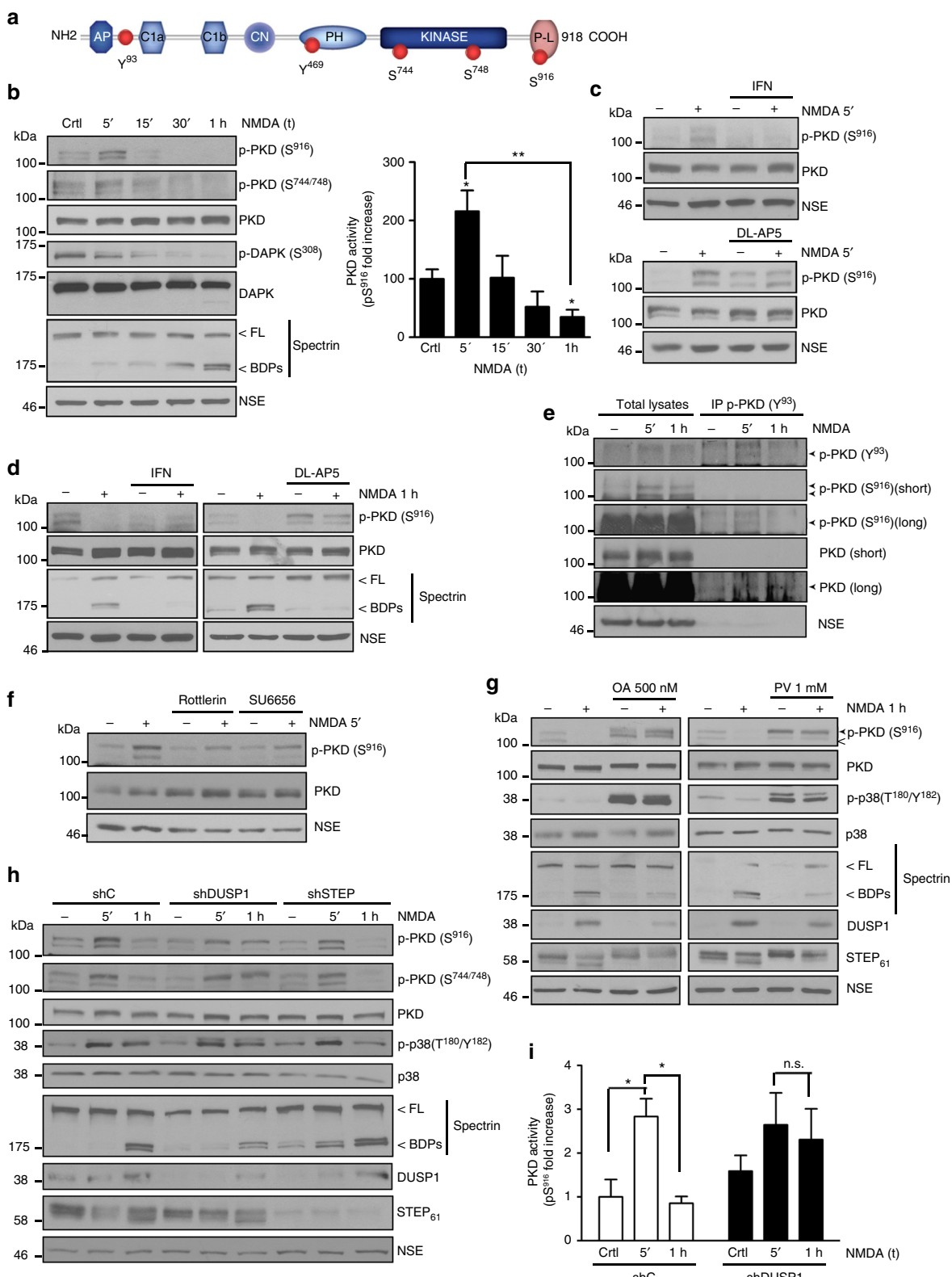

This antibody detected the upper PKD1 band in neuronal pTyr93 immunoprecipitates and its signal followed a similar pattern of phosphorylation/dephosphorylation as that for p-Ser916 following NMDA addition in the precipitated immunocomplexes (Fig. 1e). It is important to note that pTyr93 signal (and also that of pTyr469) in neurons was strongly potentiated by tyrosine phosphatase inhibitors (Supplementary Fig. 2a). These data suggest that tyrosine PPs might be highly active in mature neurons and contribute to the low level of detection of tyrosine phosphorylated residues unless potent phosphatase inhibitors are used.

Next, we analyzed the effect of the general inhibitors of PKC and Src, GF109203X (GFI) and PP2, respectively, finding that both blocked excitotoxic PKD activation (Supplementary Fig. 3a). Furthermore, rottlerin and SU6656, selective PKCδ and Src family kinase inhibitors, showed the same effect (Fig. 1f, see quantification in Supplementary Fig. 3b), indicating the specific participation of both kinases on the early PKD activation induced by NMDA.

While several kinases involved in PKD activation have been identified, the PPs mediating its inactivation remain unknown[7]. We investigated how pharmacological inhibition of different PPs could prevent excitotoxicity-induced PKD dephosphorylation. We pretreated neurons with high concentrations of the serine/threonine PPs inhibitor okadaic acid (OA, 500 nM) to inhibit PP1 and PP2A activities[31] before overstimulating NMDARs. PKD was maintained highly active in the presence of OA under basal conditions or after NMDA addition (Fig. 1g). However, low concentrations of OA (1 nM), able to block PP2A but not PP1[31], or PP2B inhibitors did not prevent PKD inactivation (Supplementary Fig. 3c, d). Treatment with the tyrosine PPs inhibitor pervanadate (PV) also increased basal PKD activity and preserved its activation after 1 h of NMDARs overstimulation (Fig. 1g). The PV-induced increase at the higher molecular weight band of p-Ser916 doublet suggested the presence of PKD1-tyrosine-hyperphosphorylated forms in this band (Fig. 1g, solid arrow head). Together, these results suggest a critical involvement of PP1 and tyrosine PPs in the regulation of neuronal PKD basal activity and in its inactivation during excitotoxicity. They also indicate that activity of PKD in mature neurons might undergo dynamic cycles of phosphorylation/dephosphorylation events due to synaptic activity, likely with the participation of GluN2A subunits and calcium. This fine dynamic synaptic modulation would explain the substantial increases in PKD phosphorylation and activity when all NMDARs or the PPs involved are blocked.

DAPK was dephosphorylated earlier than PKD (Fig. 1b), suggesting that they are regulated by different PPs. However, the kinetics of PKD dephosphorylation paralleled those of the mitogen activated protein kinase p38 (Supplementary Fig. 3e and refs. [32–34]) suggesting both could be inactivated by the same PPs. The dual specificity phosphatase-1 (DUSP1) and the brain specific striatal-enriched protein tyrosine phosphatase (STEP) are known p38 PPs activated during excitotoxicity and cerebral ischemia[35–39]. We therefore investigated DUSP1 and STEP contribution to excitotoxic PKD inactivation. STEP dephosphorylation leads to its activation detected by a mobility shift from 61 kDa ($STEP_{61}$) toward faster-migrating forms[36, 37]. NMDAR overstimulation increased DUSP1 levels and STEP activation in neurons, both changes paralleling PKD and p38 inactivation (Supplementary Fig. 3e).

Since PP1 can activate STEP downstream NMDARs overactivation[36, 40, 41], we analyzed STEP activity in the presence of OA. We observed that only PP1-inhibitory concentrations of OA blocked STEP activation and led to the accumulation of p-p38 (Fig. 1g; Supplementary Fig. 3c). PV treatment was also effective blocking STEP activation and p38 dephosphorylation (Fig. 1g). In addition, OA and PV decreased DUSP1 excitotoxic induction (Fig. 1g).

Finally, we transduced neurons with lentiviral particles encoding a control shRNA (shC) or shRNA specific for either *Dusp1* (shDUSP1) or *Step* (shSTEP) silencing. *Dusp1* knockdown significantly hampered PKD excitotoxic dephosphorylation while *Step* silencing had no effect (Fig. 1h, i). Of note, shDUSP1 exerted a very similar action to that of PV, increasing the upper band of p-Ser916 doublet, suggestive of DUSP1 acting on p-Tyr residues within active PKD1 (Fig. 1h). Furthermore, *Dusp1* but not *Step* knockdown (Fig. 1h), as PV treatment (Fig. 1g), favored the appearance of a p-p38 higher molecular weight band indicative of p38 hyperphosphorylation, indicating a similar regulation of PKD and p38 by DUSP1.

**Ischemic stroke causes neuronal inactivation of PKD.** To investigate neuronal PKD inactivation in in vivo excitotoxicity, we examined samples from cerebral ischemia. First we used a mouse model of transient cerebral ischemia that consisted in the occlusion of the middle cerebral artery (MCAO) for 1 h followed by 24 h reperfusion[42]. Hypochromatic Nissl staining marked neuronal injury in the ischemic brain, distinguishing the ischemic core at the striatum (Fig. 2a). Consistent with our in vitro results, immunofluorescence analyses of brain sections showed perinuclear and nuclear dotted staining of active PKD in neurons (NeuN+) in the striatum from sham-operated mice that was absent in the ischemic core at the equivalent region from MCAO-operated animals (Fig. 2b). In the penumbra area, a high number

**Fig. 1** PKD activity regulation in an in vitro model of NMDA-induced excitotoxicity. **a** Scheme showing activatory and autophosphorylation sites and domains in PKD1. **b** p-PKD(S916), p-PKD(S744/S748), PKD, p-DAPK(S308), DAPK, and Spectrin immunoblot analysis of primary mature cortical neurons stimulated with NMDA (50 μM) plus glycine (10 μM) (referred hereafter as NMDA) for various periods of time. Spectrin full-length (FL) and calpain-breakdown products (BDPs) are shown. (Right panel) Quantification of immunoblot signals of p-PKD(S916) relative to total PKD and the loading control neural-specific enolase (NSE). Each time point, p-PKD(S916) value was represented as fold increase relative to control untreated cultures ($n = 5$ independent experiments). **c, d** PKD, p-PKD(S916), and Spectrin immunoblot analysis of neurons pretreated for 1 h with the GluN2B-specific inhibitor ifenprodil (IFN; 10 μM) or the NMDAR antagonist DL-AP5 (200 μM), and stimulated with NMDA for 5 min (**c**) or 1 h (**d**) ($n = 3$ independent experiments). **e** Neurons were stimulated with NMDA as above for 5 min or 1 h. PKD1-Y93 phosphorylation after 5 min of NMDA treatment was detected following immunoprecipitation and immunoblotting with a novel phosphospecific monoclonal antibody. Short and long exposure images of p-PKD(S916) and PKD are included. **f** PKD, p-PKD(S916), and NSE (loading control) immunoblot analysis of neurons pre-incubated for 1 h with the PKC-δ inhibitor Rottlerin (5 μM) or the Src inhibitor SU6656 (5 μM) and treated with NMDA for 5 min ($n = 3$ independent experiments). **g** Neurons were exposed for 1 h to okadaic acid (OA; 500 nM), to inhibit serine/threonine phosphatase PP1, or pervanadate (PV; 1 mM), to inhibit tyrosine phosphatases, and then stimulated with NMDA for 1 h. Levels, processing or phosphorylation of PKD, p38, STEP, or DUSP1 were determined by immunoblot analysis ($n = 3$ independent experiments). **h** Neurons transduced with lentivirus encoding shC, shDUSP1, or shSTEP were treated with NMDA as indicated. *Dusp1* or *Step* silencing and their effect on PKD inactivation in response to excitotoxicity was analyzed by immunoblotting. **i** Quantification of immunoblot signal of p-PKD(S916) higher molecular weight band in **h** relative to total PKD and NSE, represented as fold increase relative to untreated cultures transduced with shC is shown as mean ± s.e.m. ($n = 3$ independent experiments). *$P < 0.05$, **$P < 0.01$; n.s. not significant. two-tailed unpaired Student's $t$ test. **b–h** Representative immunoblots are shown

of neurons had lost p-Ser916 signal compared to the same cortical region in a sham-operated brain (Fig. 2c). This result was reproduced in animals after MCAO for 1 h and a shorter post-ischemic time of 5 h (Supplementary Fig. 4). We further examined the potential relevance of PKD inactivation in human brain tissue from control donors and ischemic stroke patients (Supplementary Table 1). While NeuN+ cells in control cortices showed p-Ser916 immunostaining, cortices of stroke patients lacked active PKD (Fig. 2d).

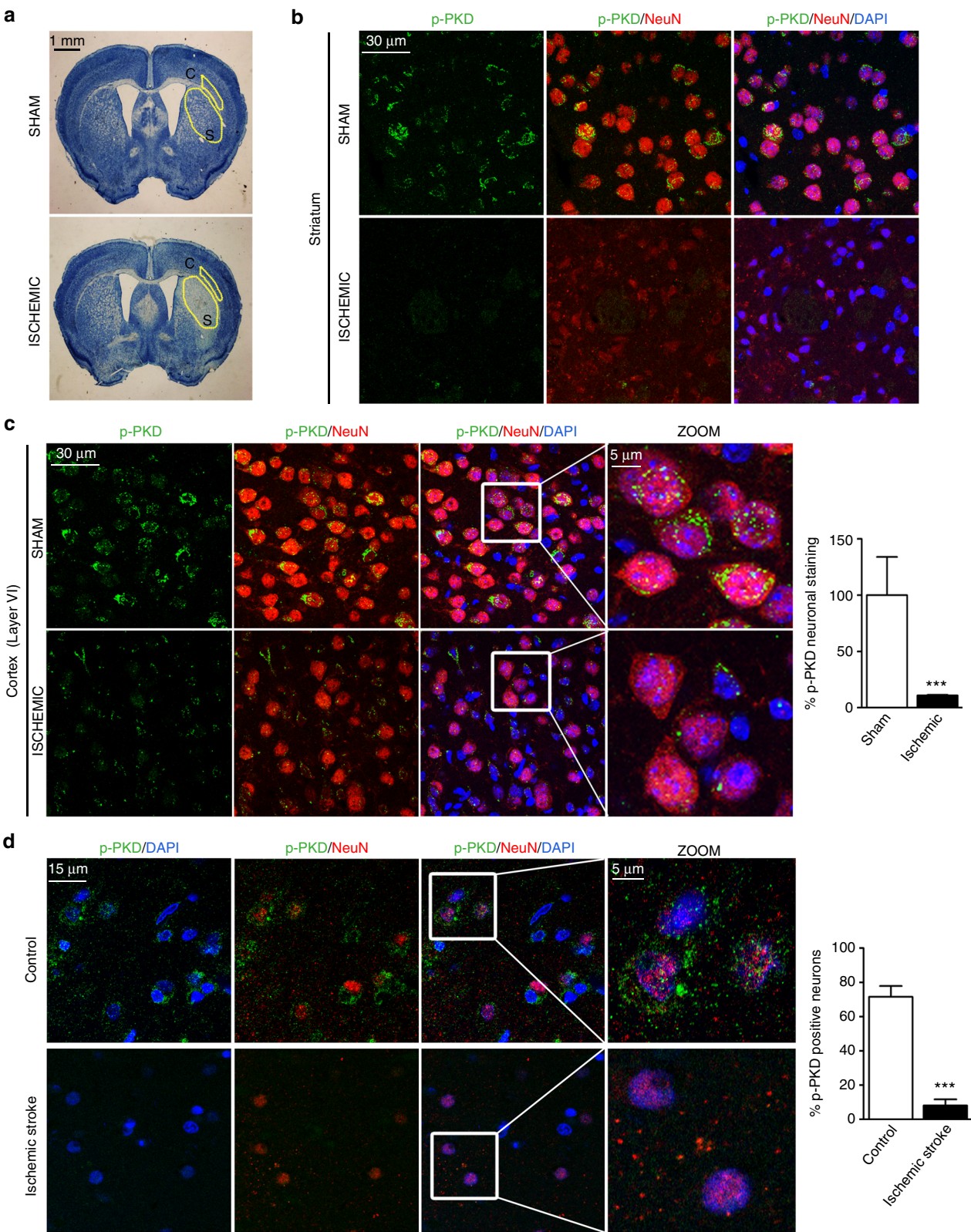

**A critical role for PKD1 in neuronal survival**. Next, we investigated whether PKD inhibition could mediate neuronal death. A dose–response and a time-course treatment of cortical neurons with the PKD-specific pharmacological inhibitor CRT0066101 (CRT)[43] showed that 10 μM CRT decreased PKD basal activity and abrogated NMDA-induced activatory response (Supplementary Fig. 5a–c). To examine the effect of this inhibitor on neuronal death, we treated cortical cultures with CRT alone or in combination with NMDA for 4 h and quantified neurons (MAP2+) bearing condensed nuclei as marker of neuronal damage. CRT alone elevated the number of neurons showing nuclear condensation and increased NMDA-induced nuclear condensation (Fig. 3a). MTT viability assays showed similar results, where preincubation with CRT decreased neuronal survival both in basal and excitotoxic conditions (77 ± 6.2% vs. 100% of viability in control cells and 36 ± 4.8% vs. 48 ± 1.9% viable neurons in NMDA-treated cultures) (Fig. 3b). No major changes were obtained at lower CRT doses, except for a small decrease in basal neuronal survival at 5 μM (Supplementary Fig. 5d).

To complete these studies, we transduced primary cortical neurons with lentiviral particles encoding two *Prkd1*-specific shRNA (shPKD1a and shPKD1b). Once confirmed PKD1 silencing (Fig. 3c), we counted living MAP+ neurons and found that *Prkd1* knockdown decreased nearly 70% the number of viable neurons (Fig. 3d). MTT assays showed that interference of *Prkd1* decreased neuronal viability in basal conditions and after NMDA-induced excitotoxicity (Fig. 3e).

We next investigated the contribution of PKD1 to neuronal survival in vivo. We generated mice with specific deletion of *Prkd1* in *CaMKIIα*-expressing neurons (PKD1 KO) by crossing *Prkd1*-floxed mice[44] (PKD1^floxed) with mice expressing Cre under neuronal *CaMKIIα* promoter (*CaMKIIα*-Cre mice) (Supplementary Fig. 6a). We confirmed Cre expression, efficient recombination in the *Prkd1* locus and specific decrease of *Prkd1* messenger RNA (mRNA) in the cerebral cortex of these mice (Supplementary Fig. 6b, c). Primary cortical neurons cultured from PKD1 KO mice showed lack of PKD1 protein compared to their control PKD1^floxed littermates (Supplementary Fig. 6d). DAB immuno-histochemistry of PKD1 KO cerebral cortex showed almost a complete disappearance of PKD antibody specific signal and a substantial decrease in p-Ser916 content (Fig. 3f). Together with results from PKD1 shRNA and immunoblot analysis with these antibodies (Supplementary Fig. 1a), these data strongly suggest that the remaining p-Ser916 signal might correspond to PKD2. Nissl staining, as well as NeuN immunofluorescence did not show noticeable differences between PKD1^floxed and PKD1 KO animals (Supplementary Fig. 6e, f). In addition, these animals did not present macroscopic differences in their cerebrovascular anatomy (Supplementary Fig. 6g). However, after experimental stroke by MCAO, PKD1 KO suffered increased neuronal injury compared to PKD1^floxed animals, as determined by Nissl staining and T2-weighted magnetic resonance imaging (MRI) (Fig. 3g). Our results suggest that although there is no neuronal loss in PKD1 KO mice, their neurons are more sensitive to excitotoxicity and oxidative stress damage. To test this notion, we first determined ROS levels in PKD1 KO vs. PKD1^floxed cultured cortical neurons untreated or treated with NMDA (Fig. 3h; Supplementary Fig. 6h). Although confocal microscopy images evidenced slight ROS increases in untreated PKD1 KO neurons (Supplementary Fig. 6h), flow cytometry quantification analysis showed that differences between the two genotypes only reached significance under excitotoxic conditions, where PKD1 KO contained higher ROS amounts (Fig. 3h). We used MDA, a marker for lipid oxidation by oxidative stress, to analyze the effects of oxidative stress in PKD1^floxed and PKD1 KO brain after sham or MCAO surgery. Importantly, immunofluorescence and quantitative analyses of brain sections showed substantially higher MDA staining in neurons (NeuN+) of sham-operated PKD1 KO compared to PKD1^floxed mice (Fig. 3i). Cerebral ischemia increased MDA labeling in both genotypes but levels in PKD1 KO registered a robust three-fold increase over those in PKD1^floxed brain (Fig. 3i).

**Regulation of IKK/NF-κB pathway during excitotoxicity**. The inactivation of PKD by excitotoxicity led us to hypothesize that the PKD1/IKK/NF-κB pathway may be constitutively active in neurons and that NMDA-induced excitotoxicity and neuronal death could be the consequence of the shut-off of this signaling cascade. The NF-κB transcription factor, formed in neurons mainly by the p65–p50 dimer[45], is sequestered in the cytosol through its binding to IκBα, which undergoes proteasomal degradation after IKK phosphorylation[46]. Therefore, fluctuations in IKK activity are usually translated into changes in the nuclear-cytosolic localization and transcriptional activity of NF-κB[46]. To challenge our hypothesis, we analyzed IKK activity during excitotoxicity finding a decrease in p-Ser176/177 signal indicative of its inactivation after 1 h of NMDAR overstimulation (Fig. 4a). Longer incubations with NMDA produced a stronger decrease in p-IKK levels paralleled by an increase in IκBα (Supplementary Fig. 7a).

We examined the localization of NF-κB p65 (referred as NF-κB) in cultured neurons observing that nearly 100% of MAP2+ neurons presented NF-κB in the nucleus (Fig. 4b) and that NMDA treatment induced significant decreases in NF-κB nuclear signal (Fig. 4b, see quantification in the right panel). NF-κB activity was also explored detecting its Ser536 phosphorylation by immunofluorescence. Phosphorylated NF-κB signal constitutively localized at neuronal nuclei and underwent a strong loss after 2 h of NMDA addition (Supplementary Fig. 7b).

To further confirm excitotoxic changes on NF-κB transcriptional activity in neurons, we performed luciferase assays and observed a significant luciferase activity decrease in NMDA-treated neurons (Fig. 4c). Finally, using RT-qPCR we also

**Fig. 2** Neuronal inactivation of PKD in mouse and human ischemic brain. **a–c** Wild-type mice were sham-operated or subjected to 60 min of MCAO and killed 24 h after reperfusion. (**a**, left panels) Representative images of coronal brain sections stained with Nissl are shown. The ischemic core in the striatum (S) and in the adjacent cortex (C) corresponding to the penumbra area and the equivalent areas in sham-operated animal are depicted. **b**, **c** Representative confocal microscopy images showing predominant localization of p-PKD(S916) staining in NeuN+ cells in brain from sham-operated animals, the absence of active kinase, and NeuN staining at the striatum (**b**) or the decrease in p-PKD(S916) signal on NeuN+ cells at the cortical penumbra area in ischemic brain (**c**, left panels). (**c**, right panel) Percentage of NeuN+ cells containing p-PKD(S916) staining in penumbra zone of MCAO-operated mice compared to the equivalent cortical region of sham-operated animals (n = 100 neurons; n = 3 sections per animal, n = 3 animals per condition). **d** Decreased number of neurons with p-PKD(S916) staining in postmortem human ischemic stroke samples compared to that from control subjects. Representative confocal microscopy images showing predominant localization of p-PKD(S916) staining in NeuN+ cells in samples C-1 and S-2 are shown (Supplementary Table 1). (Right panel) Percentage of neurons bearing active p-PKD (n = 30–50 neurons per section; n = 3 individuals per condition). **c**, **d** Zoom images from boxed regions are also shown. For quantifications, mean ± s.e.m. were derived from the indicated number of samples and analyzed with two-tailed unpaired Student's *t* test. ***P < 0.001

determined that mRNA levels of *Sod2* and those of *Bdnf*, a prosurvival neurotrophin whose transcription can be regulated by NF-κB[47], were downregulated by excitotoxicity (Fig. 4d). Together, these data indicate that neurons present a substantial basal NF-κB activity that is downregulated under excitotoxic conditions.

To determine the relevance of this pathway in vivo, we examined neuronal nuclear NF-κB localization in MCAO-operated mice. According to our in vitro data, NF-κB was present in the nucleus of almost all NeuN+ cells in the striatum and cortex from sham-operated mice. However, ischemia provoked a massive neuronal NF-κB nuclear depletion in the

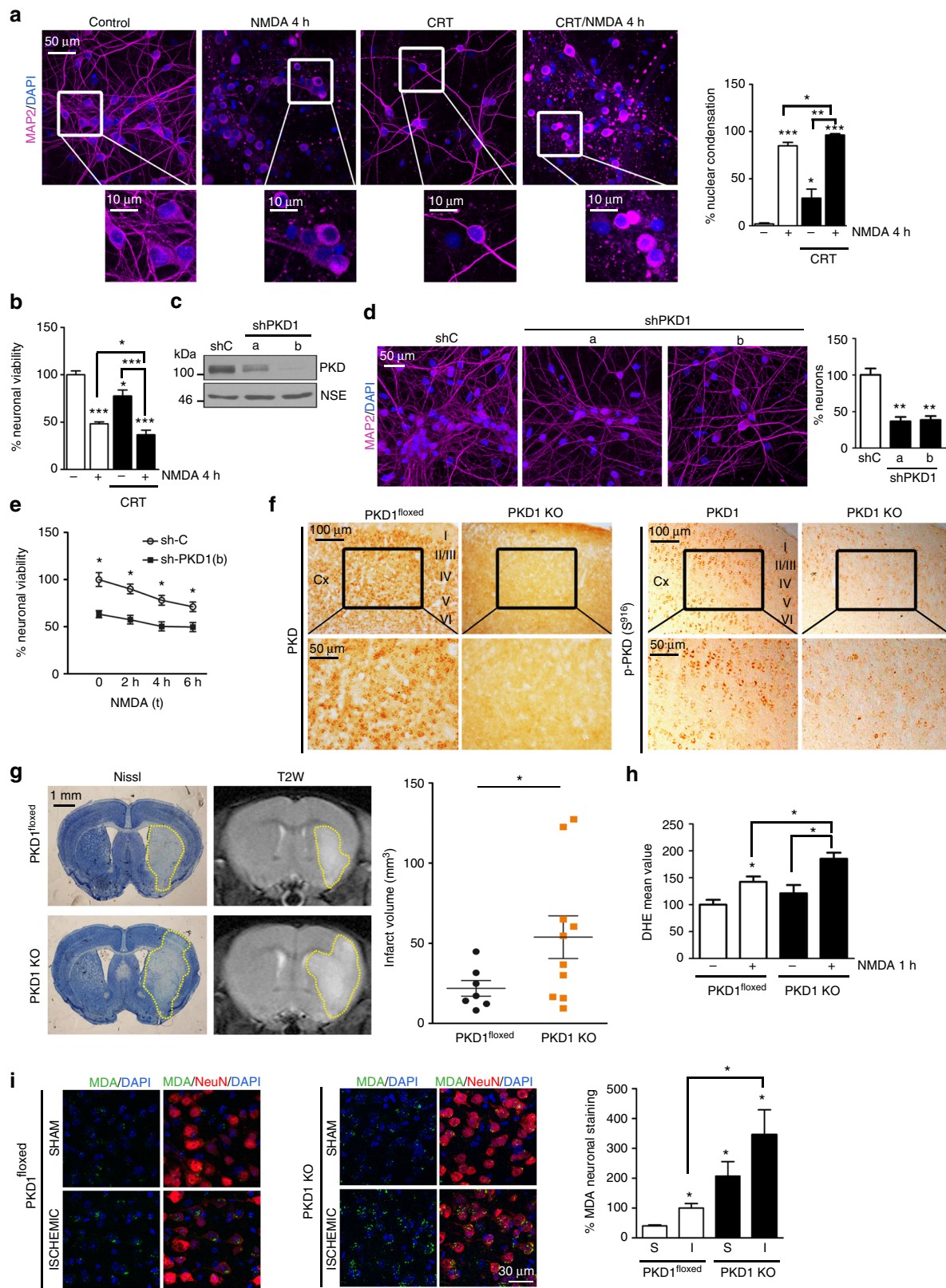

striatal ischemic core (Supplementary Fig. 7c) and a substantial nuclear loss in the cortical penumbra area (Fig. 4e). Importantly, we obtained similar results in human postmortem brain samples from ischemic stroke patients, where NF-κB was absent in neuronal nuclei from ischemic stroke patients but presented nuclear localization in control subjects (Fig. 4f).

Collectively, our results strongly suggest that the PKD1/IKK/ NF-κB pathway is constitutively "on" in healthy neurons and suffers an early and sustained "shut-off" in human brain after ischemic stroke and in experimental models of excitotoxicity and brain ischemia.

**PKD1 protects neurons through the IKK/NF-κB/SOD2 pathway.** Next we assessed the effect of a dephosphorylation-resistant and constitutively active PKD1 mutant in neuroprotection from excitotoxicity. To engineer this mutant, we substituted the four critical residues involved in PKD activation (Tyr93, Tyr469, Ser744, and Ser748) (Fig. 1a) by glutamic acid, to mimic their phosphorylation state. In vitro kinase assays confirmed its constitutive activity (Supplementary Fig. 8a). We fused this quadruple mutant to GFP in a lentiviral vector (PKD1-Ca) bearing human synapsin neurospecific promoter (SYNpr) (Fig. 5a). GFP alone expressed under the same promoter was used as control lentivirus. Neurons transduced with PKD1-Ca lentivirus expressed high amounts of active PKD1 (Fig. 5b) and presented substantial increases in IKK activity (Fig. 5b) and a highly significant induction of SOD2 levels relative to GFP-transduced cells (Fig. 5b). Accordingly, NMDA-induced ROS production in PKD1-Ca neurons was very low ($111 \pm 10.4\%$) and similar to that of untreated GFP neurons, compared to NMDA-treated GFP neurons ($148 \pm 7.8\%$) (Fig. 5c).

Analysis of MAP2 staining confirmed that PKD1 activation confers neuroprotection against NMDA-induced excitotoxicity (Fig. 5d). Most GFP neurons were not viable after 4 h of NMDA treatment, and presented disorganized MAP2 staining, whereas a prominent number of PKD1-Ca neurons remained viable at this time (Fig. 5d). Importantly, a high number of PKD1-Ca neurons survived even 24 h upon NMDA addition ($50.4 \pm 11\%$), when almost all GFP neurons were dead ($13.5 \pm 3\%$) (Fig. 5d). Furthermore, NF-κB remained in the nucleus of most PKD1-Ca transduced neurons after 1 h of NMDA treatment in contrast to its cytosolic translocation in a high proportion of GFP neurons (Fig. 5e). The possibility that PKD1-Ca increased neuronal survival by inhibiting $Ca^{2+}$ influx evoked by NMDARs overactivation was ruled out after performing $Ca^{2+}$ imaging analysis (Supplementary Fig. 8b).

To demonstrate that increases in SOD2 conferred by PKD1-Ca were dependent on IKK/NF-κB pathway, we treated transduced neurons with the IKK inhibitor SC-514. IKK inhibition decreased SOD2 levels in PKD1-Ca transduced cells (Fig. 5f, left and medium panels). The action of SC-514 was confirmed by the accumulation of IκBα (Fig. 5f, left and right panels). We finally examined the effect that pharmacological inhibition of IKK had on PKD1-mediated neuroprotection. MTT assays showed that IKK inhibition reduced the resistance of PKD1-Ca neurons to excitotoxic death: $98.6 \pm 0.6\%$ of PKD1-Ca-transduced neurons survived after 4 h of NMDA treatment, whereas only a $73.79 \pm 0.02\%$ of them were alive after combining SC-514 and NMDA treatments (Fig. 5g). Together, our results indicate that PKD1 is neuroprotective against excitotoxic insults by busting the IKK/ NF-κB/SOD2 axis and contributing to ROS detoxification.

**PKD1 protects against kainic acid-induced excitotoxicity.** To investigate PKD1 neuroprotection in vivo, we selected a model of KA-induced neurodegeneration that elicits selective excitotoxic neuronal death particularly in limbic structures (i.e., hippocampal CA1 and CA3 regions) in the rat and mouse brain[3].

We first observed in cultured primary neurons that similarly to NMDA, excitotoxic concentrations of KA produced a transient activation of PKD followed by inactivation (Fig. 6a). MTT assays and MAP2 staining showed that KA reduced neuronal viability, and that KA-induced neuronal death was mediated by KA receptors (KARs) (Supplementary Fig. 9a–c). Importantly, KA also provoked the exit of NF-κB from neuronal nuclei (Fig. 6b).

To confirm that PKD and NF-κB were inactivated in neurons during in vivo KA-induced excitotoxicity, we examined the CA1 hippocampal area in control adult rats that received an intraperitoneal injection of saline or KA. CA1 neurons from control saline-injected animals presented active PKD and nuclear NF-κB staining (Fig. 6c). In line with our observations after cerebral ischemia, KA treatment induced condensation of neuronal nuclei in the damaged CA1 area and a marked decrease of NeuN, p-PKD and nuclear NF-κB content in neurons (Fig. 6c), while p-PKD signal emerged in a different cell population. Altogether, these results strongly suggest that excitotoxicity induced by NMDARs or KARs overstimulation triggers similar molecular mechanisms that result in PKD and NF-κB neuronal inactivation preceding their death.

**Fig. 3** Elimination of PKD1 activity enhances neuronal death in vitro and in vivo. **a** (Left panels) Representative confocal microscopy images of DAPI and MAP2 staining of primary neurons treated for 1 h with CRT (10 μM) or vehicle (DMSO) followed by 4 h of NMDA. (Right panel) Percentage of these neurons bearing condensed nuclei relative to neurons treated 4 h with NMDA ($n = 120$–150 neurons per condition; $n = 3$ independent experiments). **b** Neuronal viability measured by MTT assays after treatment with CRT and NMDA as above, relative to untreated neurons (triplicates per condition, $n = 5$ independent experiments). **c** PKD and NSE (loading control) immunoblot analysis and **d** confocal microscope images of MAP2 and DAPI staining of cortical neurons transduced with lentivirus encoding shC, shPKD1a, or shPKD1b. (**d**, right panel) Quantification of MAP2+ cells relative to shC-transduced cultures ($n = 70$–200 neurons per condition; $n = 3$ independent experiments). **e** shC- or shPKD1b-transduced neuronal cultures were treated with NMDA for the indicated times and viability was evaluated by MTT assays. Data are expressed relative to shC-transduced untreated neurons (triplicates per condition, $n = 4$ independent experiments). **f** PKD (left panels) and p-PKD(S916) (right panels) DAB immunostaining in cortex sections from PKD1[floxed] and PKD1 KO mice. Representative images and zoom-boxed regions are also shown. **g** PKD1[floxed] and PKD1 KO mice were subjected to 60 min of MCAO. Representative images of (left panels) Nissl-stained coronal brain sections and (middle panels) coronal T2-weighted MRI images obtained from the same animals 24 h after reperfusion. (Right panel) quantification of the infarct volume ($mm^3$) in PKD1 KO ($n = 10$) and PKD1[floxed] mice ($n = 7$). Yellow dotted lines mark the ischemia boundary. **h** Flow cytometry analysis of DHE staining in PKD1[floxed] and PKD1 KO cultured neurons unstimulated or after NMDARs overstimulation for 1 h. Data are shown relative to untreated PKD1[floxed] neurons ($n = 4$ independent experiments). **i** Representative confocal microscopy images of MDA-, NeuN-, and DAPI-stained cortical penumbra area from MCAO-operated mice and the equivalent cortical region of sham-operated animals in PKD1[floxed] and PKD1 KO. (**i**, right panel) Percentage of NeuN+ cells containing MDA staining in penumbra zone of MCAO-operated mice (I) compared to the equivalent cortical region of sham-operated animals (S) expressed relative to that of sham PKD1[floxed] ($n = 150$ neurons; $n = 2$ sections per animal, $n = 3$ animals per condition). For quantifications, mean ± s.e.m. were derived from the indicated number of independent experiments. *$P < 0.05$, **$P < 0.01$, ***$P < 0.001$; two-tailed unpaired Student's t test (**a–f**, **h–i**) and two-tailed unpaired Student's t test with Welch's correction (**g**)

Next we assayed the effects of PKD1-Ca neurospecific expression in neuroprotection from KA-induced death. This was initially performed in vitro, analyzing neuronal cultures transduced with GFP or PKD1-Ca lentivirus and treated with KA for 48 h. We found almost complete lack of neuronal death in PKD1-Ca cultures, in contrast to the extensive neuronal death provoked by KA in GFP neurons, where nearly a 60% of the neuronal population died (Fig. 6d).

Finally, to investigate PKD1 neuroprotection in vivo, we stereotaxically injected GFP or PKD1-Ca lentiviral particles into

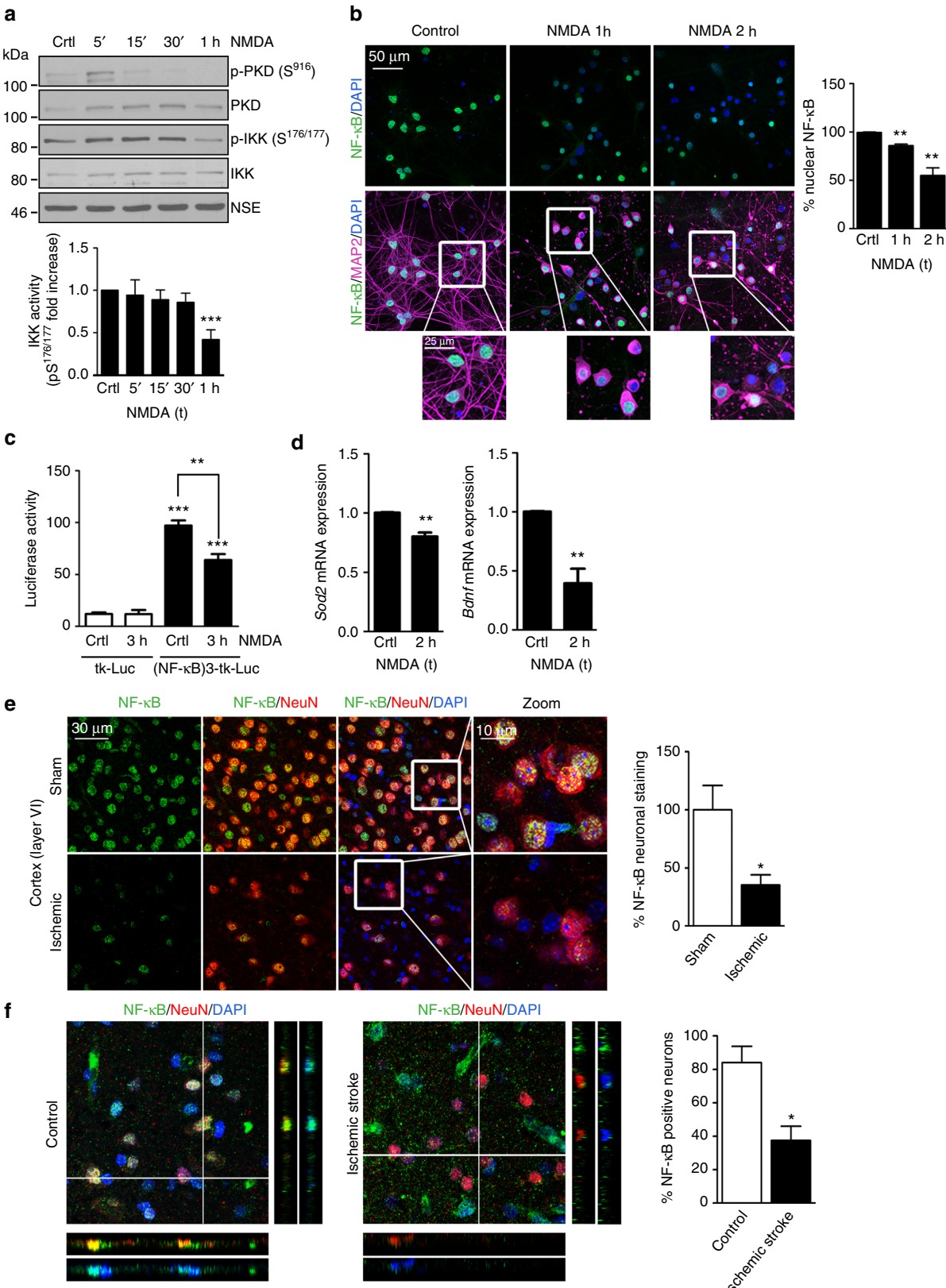

the right and left CA1 region of the rat hippocampus, respectively, previous to KA administration (Fig. 7a). In saline-injected animals, GFP+ neurons in CA1 presented nuclear localization of NF-κB independently of whether they were transduced (GFP+) or not (Fig. 7b, see details in zoom images). By contrast, after KA injection there was a clear damage and loss of transduced neurons in the right CA1 GFP-transduced side, as evidenced by spare fragmented GFP staining and condensation of NeuN-stained nuclei (Fig. 7c). Remaining neurons looked damaged, presented aberrant shape and smaller compacted nuclei, and sparse nuclear NF-κB staining (Fig. 7c). However, neurons in the left CA1 side appeared healthier, not only those transduced with PKD1-Ca (GFP+), but also non-transduced neurons surrounding them, many of which still contained NF-κB in their nuclei (Fig. 7c).

To further support PKD1-mediated neuroprotection using quantitative approaches, we measured hippocampal CA1 neuronal degeneration by FluoroJade B staining. We found a highly significant decrease in KA-induced neuronal death (represented as the % of FluoroJade B staining), in the left CA1 side transduced with PKD1-Ca compared with the right side GFP-transduced (12.65 ± 3% PKD1-Ca vs. 28.19 ± 3.6% GFP) (Fig. 7d).

## Discussion

Neurons are considered to be highly sensitive to oxidative stress damage[48], but precisely because they are post-mitotic cells that survive for many years, they must have well developed natural defences against oxidative stress. Here we have discovered that a constitutively active neuronal PKD1/IKK/NF-κB/SOD2 oxidative stress-detoxifying cascade is turned off during excitotoxic neurodegeneration, depleting neurons of this antioxidant natural defence and greatly hampering their survival (Fig. 7e). Our results support the notion that PKD1 enhances IKK activity, NF-κB nuclear localization and SOD2 expression, and decreases ROS production in neurons under oxidative stress conditions.

Excitotoxic damage accumulated along aging or enhanced by acute or chronic neurodegenerative disorders could gradually deteriorate this antioxidant pathway. PKD1 activity withdrawal, accompanied by NF-κB nuclear exit, and subsequent decrease in SOD2 levels would decrease the threshold of the amount of ROS neurons can cope with. ROS elimination through the action of SOD2 would be severely hampered (Fig. 7e). Cultured $Prkd1^{-/-}$ immortalized fibroblasts showed increased ROS production due to a decrease in the threshold of mitochondrial depolarization[49], suggesting the possibility that this mechanism may also contribute to increase ROS levels in neurons. The accumulation of

ROS in PKD1-deficient neurons, by lack of clearance or by overproduction, may increase oxidative damage and neuronal vulnerability and death. Accordingly, we observed increased MDA labeling in the brain of PKD1 KO mice relative to PKD1$^{floxed}$ littermates.

This PKD–ROS detoxification pathway was identified in a pancreatic cancer mouse model and in cancer cell lines, having as a result the promotion of precancerous lesions[50] and the potentiation of cellular survival[12], respectively. Contrary to tumor cells, this activation is not sustained in neurons, as demonstrated by decreases in PKD and IKK activities and in NF-κB nuclear localization and activity preceding excitotoxic neuronal death. A transient activation of PKD was found also in dopaminergic cells treated in vitro with hydrogen peroxide, but mechanisms of inactivation or downstream signaling cascades were not explored[51]. Here, we identify for the first time a molecular mechanism involved in PKD dephosphorylation downstream overstimulation of endogenous NMDARs. This inactivation mechanism depends on PP1 and DUSP1, two PPs whose activities are increased under excitotoxicity[35, 40] (Fig. 7e).

A key point in PKD sustained activation and survival of cancer cells under oxidative stress might relay on their deficiencies in the control of PKD inactivation mechanisms achieved by PPs. For instance, DUSP1 levels inversely correlate with NF-κB activity and their malignancy grade in prostate cancer[52]. In this context, while specific inhibition of PKD1 could provide clear therapeutic advantages in cancer, our results indicate that this type of treatments could represent a high risk for associated neurodegeneration. Conversely, the future development and therapeutic application of PKD activators for neuroprotection should be exquisitely targeted to neurons to unequivocally avoid undesired pro-survival effects on other cell types.

Regarding the existing controversy on NF-κB contribution to neuroprotection and neurotoxicity[18], our data highlight the importance of this pathway for neuronal survival. This study strengthens the need to perform careful cell-type specific analysis of this and other signaling pathways in complex tissues, identifying pathological changes occurring in cells subpopulations. Revisiting published results obtained from tissue homogenates is essential before drawing final conclusions of the contribution of a specific pathway to disease.

In summary, this study constitutes the first one demonstrating that PKD1 confers neuroprotection against the oxidative stress produced after overstimulation of endogenous glutamate receptors by triggering antioxidant defences and promoting neuronal survival in an excitotoxic environment. On the basis of our

**Fig. 4** Downregulation of IKK/NF-κB pathway in neurons during in vitro and in vivo excitotoxicity parallels PKD inactivation. **a** (Top panel) Representative p-PKD(S916), PKD, p-IKK(S176/S177), IKK, and NSE (loading control) immunoblot analysis of cortical neurons stimulated with NMDA for the indicated times and (bottom panel) quantification of IKK activity after normalizing p-IKK(Ser176/177) densitometric values with those of total IKK and NSE. For each time point, p-IKK values were represented as fold increase relative to untreated cultures ($n = 5$ independent experiments). **b** Representative images of MAP2, NF-κB, and DAPI staining of cortical neurons treated with NMDA for 1 or 2 h. Zoom-boxed regions are also shown. (Right panel) Number of neurons bearing NF-κB nuclear staining before and after NMDARs overstimulation relative to untreated cells ($n = 50$–$100$ neurons per condition; $n = 3$ independent experiments). **c** Cortical neurons were transfected with tk-Luc and (NF-κB)3-tk-Luc reporter vectors. Luciferase activity was measured after 48 h in unstimulated neurons or stimulated with NMDA 3 h. Firefly luciferase activity was normalized to that of Renilla luciferase activity in each sample, and expressed relative to that of tk-Luc control ($n = 3$ independent experiments). **d** RT-qPCR analysis of $Sod2$ and $Bdnf$ mRNA levels in cortical neurons untreated or treated with NMDA for 2 h shown relative to untreated neurons ($n = 3$ independent experiments). **e** Representative confocal images of NeuN, NF-κB, and DAPI staining in cortex of sham- or MCAO-operated animals. Zoom-boxed regions are also shown. (Right panel) Percentage of NF-κB nuclear staining in the penumbra cortical zone from MCAO brain relative to the equivalent region from sham-operated animals ($n = 100$ neurons, $n = 3$ sections per animal, $n = 3$ animals per condition). **f** Representative images of maximal projections and orthogonal views of a z-stack for double NF-κB/NeuN and NF-κB/DAPI staining of postmortem brain samples from control subjects (left panels) and ischemic stroke patients (medium panels). Images correspond to samples C-1 and S-2 (Supplementary Table 1). (Right panel) Percentage of neurons bearing nuclear NF-κB ($n = 30$–$50$ neurons per section; $n = 3$ individuals per condition). For quantifications, mean ± s.e.m. were derived from the indicated number of independent experiments or samples and analyzed with two-tailed unpaired Student's $t$ test. *$P < 0.05$, **$P < 0.01$, ***$P < 0.001$

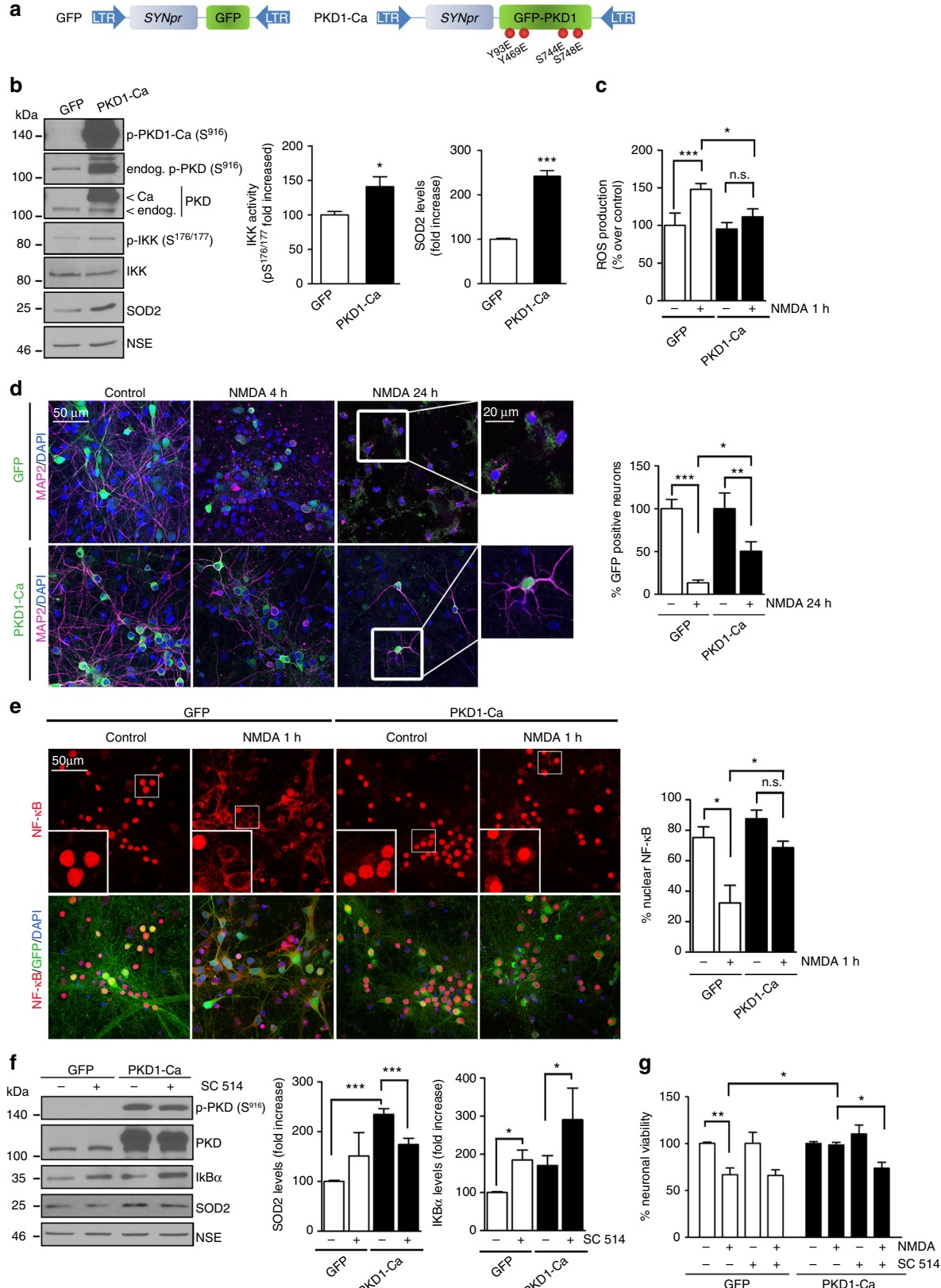

findings, we propose that the development of a broad therapeutic strategy based on preserving PKD1 activity in neurons might be beneficial to slow-down neuronal loss taking place during aging or in a broad range of acute and chronic neurodegenerative diseases by enhancing neuronal natural antioxidant defences.

## Methods

**Materials and chemicals**. NMDA, glycine, cytosine β-ᴅ-arabinofuranoside (AraC), poly-ʟ-lysine, ʟ-laminin, IFN, [3-(4,5-dimethylthiazol-2-yl)-2,5-diphe-nyltetrazolium bromide] (MTT), Rottlerin, SU6656, EGTA, Z-VAD-FMK, dime-thylsulfoxide (DMSO), and sodium orthovanadate (Pervanadate, PV) were from Sigma-Aldrich (St Louis, MO, USA). Bisindolylmaleimide I (GFI), Ca²⁺ ionophore

A23187, and SC-514 were from Calbiochem (UK). Okadaic acid (OA), 3-(4-chlorophenyl)-1-(1,1-dimethylethyl) 1-H-pyrazolo[3,4-d] pyrimidin-4-amine (PP2), DL-2-amino-5-phosphonopentanoic acid (DL-AP5), and 6-cyano-7-nitro-quinoxaline-2, 3-dione (CNQX) were from Tocris Bioscience (Bristol, UK). Cyclosporine A (CsA) and FK-506 were from Sandoz (Vienna, Austria) and LC Laboratories (Wobum, MA, USA), respectively. KA was from Abcam (Cambridge, UK). CRT0066101 was kindly provided by Cancer Research UK Transfer Technology Department (London, UK).

**Commercial antibodies.** Rabbit polyclonal antibodies were: PKD, phospho-PKD-S$^{916}$ and phospho-PKD-S$^{744/748}$, p38 and phospho-p38-T$^{180}$/Y$^{182}$, IKKβ, NF-κB phospho-p65-S$^{536}$ (Cell Signaling Technology, Beverly, MA, USA); phospho-S$^{176/177}$ IKKα/β (Biorbyt, San Francisco, CA, USA); NSE (ICN Biomedicals; Costa Mesa, CA, USA); total and phospho-S$^{308}$ DAPK (Sigma-Aldrich); Superoxide dismutase 2 (SOD2), microtubule-associated protein 2 (MAP2), PKD/PKC mu (phospho-Y463) (Abcam); GFP (ThermoFisher Scientific Invitrogen, Waltham, MA, USA); NF-κB p65, IκBα, DUSP1 (Santa Cruz Biotechnology, CA, USA). Mouse monoclonals were: NeuN, Spectrin (Millipore Corporation, Billerica, MA, USA), STEP (Novus Biologicals, Littleton, CO, USA), MDA (Jaica, Japan). Detailed information about all the above-mentioned antibodies and dilutions used for the different applications is given in Supplementary Table 2. Horseradish peroxidase-conjugated anti-rabbit and anti-mouse secondary antibodies were from Santa Cruz Biotechnology and Alexa-Fluor-488, -555, and -647 conjugated antibodies were from ThermoFisher Scientific.

**Generation of phosphospecific PKD1 p-Tyr93 monoclonal antibody.** Eight-week-old female BALB/c mice produced at the animal care facility at Centro Nacional de Biotecnología (CNB, CSIC, Madrid, Spain) were immunized subcutaneously with a KLH-conjugated synthetic phosphopeptide corresponding to amino acids 88–98 of human PKD1 (homologous to mouse amino acids 86–96, containing Y93) (KFPE*CGFpYGMY), where *C refers to acetamidomethyl (Acm) cysteine. Spleen cells from these mice were then fused with P3X63Ag8.653 (ATCC CRL-1580, mycoplasma free) mouse myeloma cells. Hybridoma supernatants were screened for the presence of phosphospecific antibodies by enzyme-linked immunosorbent assay (ELISA) using phosphopeptide and nonphosphopeptide bound to the plate as described previously[53]. Selected hybridomas, producing antibodies that reacted specifically with the phosphopeptide, were cloned twice by limiting dilution. We selected a hybridoma whose supernatant reacted specifically with the phosphopeptide and recognized specifically PKD1 phosphorylated at Tyr93 in rat neuronal lysates after pervanadate (PV; 1 mM) or H$_2$O$_2$ (1 mM) stimulation (Supplementary Fig. 2). This antibody was named: Phospho-PKD1 (Tyr93) monoclonal antibody (clone 6E8). Its isotype was determined by ELISA as IgG2b.

**Experimental animals.** Mice with neuronal conditional deletion of *Prkd1* (PKD1 KO mice) in a C57BL/6 background were obtained after crossing *Prkd1*$^{loxP/loxP}$ animals (PKD1$^{floxed}$)[44] with *CaMKIIα*-Cre mice (stock number 005359, The Jackson Laboratory, Bar Harbor, ME, USA). Only male PKD1$^{floxed}$ and PKD1-KO mice were used and littermates were employed in each independent experiment for comparison purposes. Genotyping and recombination analysis in mouse brain cortex was performed by PCR using specific pairs of primers[44]. Presence of Cre and flox cassettes was also tested.

Wild-type C57BL/6, C57BL/6 PKD1$^{floxed}$, and PKD1-KO embryos of 17 days and Wistar rat embryos of 19 days were produced at the animal care facility at Instituto de Investigaciones Biomédicas "Alberto Sols" (IIBm, CSIC-UAM, Madrid, Spain). Adult male Wistar rats (8–10 weeks old) weighing 250–280 g were used for

KA treatment. Rats were obtained from the animal care facility at Centro de Biología Molecular "Severo Ochoa" (CBMSO, CSIC-UAM, Madrid, Spain).

All animals were maintained under 12-h light/12-h dark cycle and with access to food and water ad libitum. Procedures involving animals had been approved by institutional (Consejo Superior de Investigaciones Científicas and Universidad Autónoma de Madrid) and local (Instituto de Investigaciones Biomédicas "Alberto Sols", Centro de Biología Molecular "Severo Ochoa", and Centro Nacional de Biotecnología) Ethical Committees, and were conformed to the appropriate national legislations (RD 53/2013) and the guidelines of the European Commission for the accommodation and care of laboratory animals (revised in Appendix A of the Council of Europe Convention ETS123).

**Transient cerebral ischemia mouse model.** Transient focal ischemia was produced in 4-month-old male C57Bl/6 mice (30 g body weight) by intraluminal occlusion of the middle cerebral artery for 60 min followed by reperfusion (procedure referred as "MCAO"), as described[42]. Briefly, mice were anesthetized with 2.5% isofluorane before a 1.1-cm length of 7-0 monofilament silk suture was introduced into the common carotid artery up to the level where the middle cerebral artery branches out; animals were sutured then and placed in their cages with free access to water and food. After 60 min, animals were reanesthetized and the filament was removed to allow reperfusion. Anesthesia was discontinued again, and the animals were allowed to recover. Mice were reanesthetized and examined 5 h or 1 day after reperfusion with T2W MRI, at the Magnetic Resonance facility (Instituto de Investigaciones Biomédicas "Alberto Sols"), to evaluate the extent of edema and brain infarction. To calculate the total infarct volume in mm$^3$, the infarct area was measured in all the series of images obtained in MRI-T2W for each animal. MRI-ADC images were also acquired after 2 h of reperfusion in the groups of animals subjected to the short MCAO procedure to identify and quantify edema within the ischemic hemisphere.

**Human brain samples.** Brain samples from ischemic stroke patients ($n = 4$), and control subjects ($n = 3$) and postmortem intervals between 3 and 15 h were obtained from the Institute of Neuropathology Brain Bank (for details see Supplementary Table 1) following Spanish legislation and local Ethics Committee guidelines and processed as described[54]. Briefly, one brain hemisphere was immediately cut in coronal sections (1 cm-thick) and selected areas were rapidly dissected, frozen over dry ice on metal plates, placed in individual plastic bags, and stored at −80 °C for their use in biochemical studies. The other hemisphere was used for morphological studies after fixation by immersion in 4% buffered formalin for 3 weeks. Samples of selected regions of the brain were embedded in paraffin and 15-μm sections were stained for neuropathological studies with hematoxylin and eosin, periodic acid-Schiff (PAS) and Klüver–Barrera, or processed for immunohistochemistry and immunofluorescence (see "Immunohistochemistry of rat, mouse, and human brain samples" section and details for antibodies and dilutions used in Supplementary Table 2). All participants gave their written consent, and the study was approved by the local Ethics Committee (Bellvitge University Hospital-Bellvitge Biomedical Research Institute, IDIBELL), following the ethical standards recommended by the Helsinki Declaration.

**Stereotaxic intracerebral injection of lentiviral particles.** Male rats 8–10 weeks old were anesthetized with isoflurane and placed in a stereotaxic frame. Coordinates (mm) relative to bregma in the antero-posterior (AP), medio-lateral (ML), and dorso-ventral (DV) axes were as follows: CA1 (−4 AP; ±2.40 ML and −2.5 DV). GFP and PKD1-Ca lentiviral particles were injected in CA1 hippocampal regions at the right and left hemisphere, respectively, of the same animal using glass

**Fig. 5** Phosphatase-resistant active PKD1 enhances IKK/NF-κB/SOD2 pathway and reduces excitotoxicity-induced ROS levels and neuronal death. **a** Scheme of the lentiviral vector used for neuronal expression of a phosphatase-resistant active PKD1 mutant (PKD1-Ca), where glutamic acids substitute the four activatory residues (Tyr93, Tyr469, Ser744, and Ser748). GFP alone or fused to PKD1-Ca was cloned under the neurospecific human synapsin promoter (*SYNpr*). **b** Immunoblot of PKD and IKK activities and SOD2 levels from primary neurons transduced with GFP or PKD1-Ca lentivirus. Quantitative analysis of (medium panel) p-IKK(S176/177) after normalization with total IKK and NSE, or (right panel) SOD2 after normalization with NSE, in GFP-transduced neurons ($n = 3$ independent experiments). **c** ROS production in GFP- or PKD1-Ca-transduced neurons unstimulated or after NMDARs overstimulation for 1 h determined by Cell Rox deep red reagent and FACs analysis, expressed relative to untreated GFP-transduced neurons ($n = 4$ independent experiments). **d** Representative images of GFP, MAP2, and DAPI signal of GFP- or PKD1-Ca-transduced neurons treated with NMDA for 4 or 24 h. Zoom-boxed regions are also shown. Transduced neurons were GFP+. (Right panel) Percentage of GFP+ surviving neurons (transduced with GFP or PKD1-Ca) after 24 h NMDA treatment, relative to total GFP+ neurons in untreated conditions ($n = 120$–250 neurons per condition; $n = 4$ per group). **e** Representative images of MAP2, NF-κB, and DAPI staining of GFP- or PKD1-Ca-transduced neurons treated with NMDA for 1 h. See inserts with zoom images. (Right panel) Percentage of neurons bearing nuclear NF-κB before or after NMDARs overstimulation ($n = 50$–100 neurons per condition; $n = 3$ independent experiments). **f** GFP- or PKD1-Ca-transduced neurons were treated for 6 h with the IKK inhibitor SC-514 (10 μM) or with vehicle (DMSO). Representative immunoblots with the indicated antibodies (left panel) and quantitative analysis of (middle panel) SOD2 and (right panel) IκBα levels normalized to NSE. Data are expressed relative to control GFP+ neurons ($n = 4$ per group). **g** Neuronal viability was measured by MTT assays in GFP- or PKD1-Ca-transduced cortical cultures treated for 6 h with SC-514 or vehicle (DMSO) followed by 4 h with NMDA. Data are expressed relative to untreated neurons (triplicates per condition; $n = 4$ independent experiments). For quantifications, mean ± s.e.m. were derived from the indicated number of independent experiments. *$P < 0.05$, **$P < 0.01$, ***$P < 0.001$, n.s. not significant. two-tailed unpaired Student's $t$ test

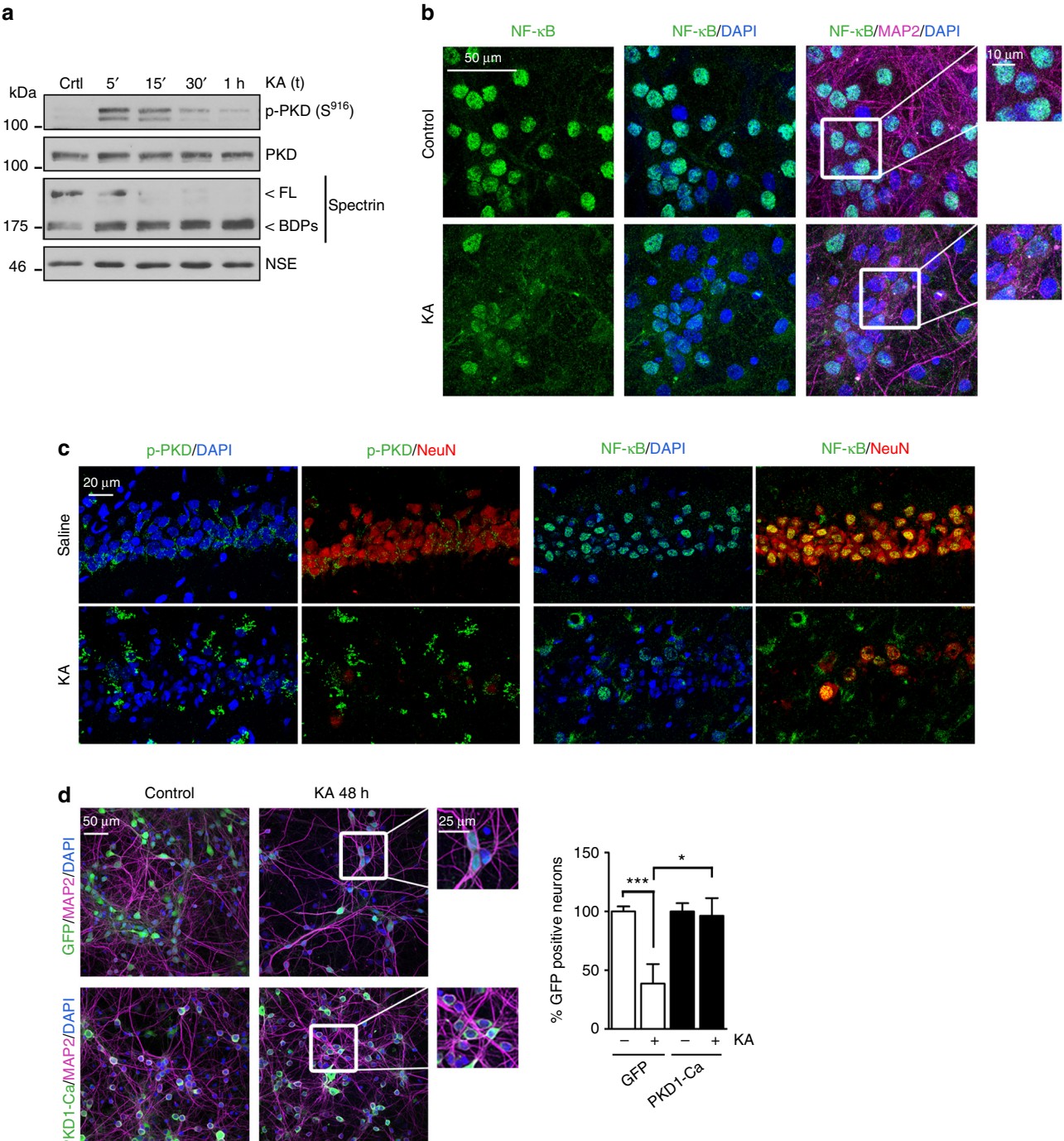

**Fig. 6** PKD1 mediates neuroprotection from in vitro KA-induced excitotoxicicity. **a** Primary cortical neurons were stimulated with excitotoxic concentrations of KA (50 μM) for various periods of time and PKD activity and Spectrin processing were determined by immunoblotting. Representative immunoblots are shown ($n = 3$ independent experiments). **b** Representative images of MAP2, NF-κB, and DAPI staining of neurons treated with KA for 4 h. Zoom-boxed regions are also shown. ($n = 3$ independent experiments). **c** Brain slices from adult male Wistar rats treated with saline or KA (8.5 mg per kg, i.p.) were stained with p-PKD(S916) or NF-κB in combination with NeuN and DAPI. Representative confocal microscopy images of the hippocampal CA1 region are shown ($n = 3$ animals per conditions). **d** Representative images of GFP fluorescence and MAP2 and DAPI staining of primary cultured neurons transduced with GFP or PKD1-Ca lentivirus and treated with KA for 48 h. Zoom-boxed regions are also shown. (Right panel) Neuronal survival was determined counting the number of living GFP+ neurons 48 h after KA treatment and shown relative to the number of GFP+ neurons transduced with GFP or PKD1-Ca in untreated conditions ($n = 200$–$400$ neurons per condition; $n = 3$ independent experiments). Data are expressed as mean + s.e.m. analyzed with two-tailed unpaired Student's $t$ test. *$P < 0.05$, ***$P < 0.001$

micropipettes. A total volume of 2.5 µl per CA1 region of each viral suspension at a concentration of $10^7$–$10^8$ pfu per ml) was infused at 0.2 µl per min using an automatic infusion pump.

**Treatment with kainic acid.** Twelve days after lentiviral CA1 intracerebral injection, rats were treated with a single intraperitoneal (i.p.) dose of KA (8.5 mg per Kg) or with an equivalent volume of 0.9% saline solution. To verify neurotoxicity induction by KA, behavioral observations were made during 3 h post injection. All KA-treated animal displayed general limbic seizure activity. Seizures were scored as previously described[55]. Three days after KA treatment, animals were anesthetized and perfused for further immunohistochemistry analysis.

**Primary culture and treatment of cortical neurons.** Rat cortical neurons were prepared from cerebral cortex of 19-day-old Wistar rat embryos as previously described[23, 24]. Briefly, cerebral cortices were dissected and mechanically

dissociated in 4 ml of MEM culture medium (Eagle's minimum medium, ThermoFisher Scientific Gibco), supplemented with 5% fetal bovine serum, 5% horse serum, 22.2 mM glucose, 0.1 mM Glutamax-I (ThermoFisher Scientific Gibco), penicillin (100 U per ml) and streptomycin (100 U per ml). Cells were plated at a density of $2.5 \times 10^5$ cells per $cm^2$ in the same medium in plates or coverslips previously treated with poly-L-lysine (100 µg per ml) and laminin (4 µg per ml). At day 7, cytosine β-D-arabinofuranoside (10 µM) was added to the culture and maintained until the end of experiments to inhibit growth of glial cells. After 14 days in vitro (DIV), neurons were pretreated or treated for different time periods as indicated with the following concentrations of compounds: 50 µM NMDA and 10 µM glycine (combination referred as "NMDA" along the manuscript), 200 µM DL-AP5, 10 µM IFN, 4 µM A23187, 2 mM EGTA, 25 µM Z-VAD-FMK, 20 µM CNQX, 3.5 µM GFI, 5 µM Rottlerin, 5 µM PP2, 5 µM SU6656, 500 nM or 1 nM OA, 1 mM PV, 200 ng per ml FK506, 100 ng per ml CsA, 10 µM CRT, 20 µM SC-514, 50 µM KA, 1 mM $H_2O_2$. Excitotoxicity was induced in cortical neurons by overstimulation of NMDARs with the co-agonists NMDA and glycine

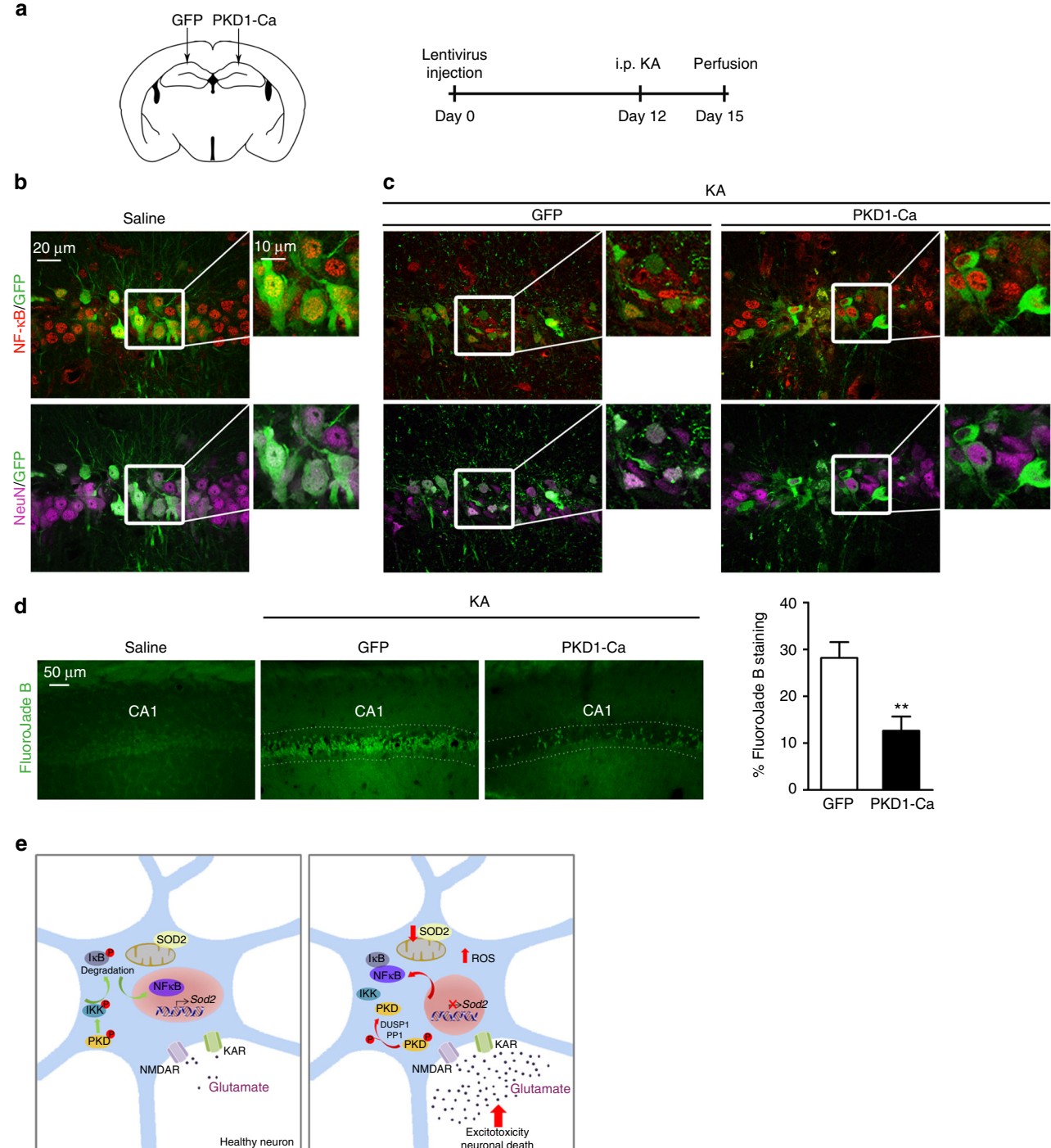

or by overstimulation of KARs with KA. Unless otherwise stated, inhibitors were added 1 h before NMDA treatment and remained in the culture media for the duration of the experiment.

Cultures of mouse cortical neurons from E17 brain cortex of PKD1[floxed] and PKD1-KO animals were prepared as previously described[56]. Briefly, tissue was incubated in a 0.25% trypsin solution in $Ca^{2+}/Mg^{2+}$ free Hank's buffered salt solution (HBSS) and dissociated using fire polished Pasteur pipettes. Then neurons were seeded at a density of $0.5 \times 10^5$ cells per $cm^2$ on coverslips or dishes as above in MEM complemented with 10% horse serum, 0.6% glucose, 0.1 mM Glutamax-I, penicillin (100 U per ml), and streptomycin (100 U per ml), for 2 h until they attached. Medium was then replaced by Neurobasal medium containing B27 supplement and 2 mM Glutamax-I (ThermoFisher Scientific Gibco). Cytosine β-D-arabinofuranoside (10 μM) was added to the culture at DIV3. Neurons were maintained in the same original medium and used at DIV11. All cultures were incubated at 37 °C in a humidified atmosphere containing 5% $CO_2$.

**Preparation of protein extracts, immunoprecipitation, and immunoblot analysis.** Protein extracts from primary cultures were prepared in RIPA as previously described[23, 24]. Briefly, cultured neurons were lysed in RIPA buffer (25 mM Tris-HCl, pH 7.6, 1% Triton X-100, 0.5% sodium deoxycholate, 0.1% SDS, 150 mM NaCl) with protease and phosphatase inhibitors for 30 min at 4 °C. Lysates were centrifuged for 30 min at 14,000 rpm at 4 °C, and resulting supernatant was considered the total lysate soluble fraction. For immunoprecipitation of PKD p-Tyr93, 1 mg of neuronal protein extracts was incubated with 50 μl of the p-Tyr93 hybridoma supernatant. After 4 h at 4 °C, immunocomplexes were bound to protein G Sepharose for 1 h at 4 °C. Beads were washed four times with RIPA buffer before solubilization in sample buffer. Equal amounts of total lysates or immunoprecipitates were resolved in SDS–PAGE and analyzed by immunoblot. Membranes were incubated with different primary and secondary antibodies and immunoreactive bands were detected by ECL (PerkinElmer, Waltham, MA USA). Immunoblot images have been cropped for presentation. Full size images are presented in Supplementary Figs. 10–15.

**RNA isolation and reverse transcription-PCR and quantitative real-time PCR analysis.** Total RNA from the cortex of PKD1[floxed] and PKD1-KO mice was isolated with RNAasy Mini Kit (Qiagen, The Netherlands), treated with RQ1 RNase-free DNase (1 U per μg RNA; Promega, Madison, WI), and reverse-transcribed into complimentary DNA (cDNA), using oligo-dT extension with Superscript II (Invitrogen, Carlsbad, CA). For PCR amplification, pairs of primers specific for mouse *Prkd1*, *Prkd2*, and *Prkd3* were used (*Prkd1*, forward 5′-GCCAAGGCCTTAAATGTGAA-3′, reverse 5′-GGAGCTTGTCGAGCTGAATC-3′; *Prkd2*, forward 5′-CTTGGATCTCCAGTCGCCGC-3′, reverse 5′-ACAGAACCACCTCCACCAGGTCA-3′; *Prkd3*, forward 5′-TGGTAATGTG CAGGGTCAAA-3′, reverse 5′-GGCTCCTCTGAGTCATCCAA-3′) and PCR products were analyzed in agarose gels.

For quantitative real-time PCR (RT-qPCR), isolation of RNA and cDNA preparation was performed as above. PCR reactions (25 μl) contained 20 ng of cDNA, 0.25 μM amplification primers, and 12.5 μl of 2× SYBR Green Mastermix (Applied Biosystems, Foster City, CA). RT-qPCR was performed in a 7900 HT FAST Real Time PCR System thermocycler (Applied Biosystems). Denaturation at 95 °C for 10 min was followed by 40 cycles of 15 s at 95 °C, and 1 min at 60 °C. The pairs of primers specific for rat transcripts are 5′-AACTGTCACAAACGGCTGTGC-3′ and 5′-TTGTCATCGCTCCCTTCTTC-3′ to *Prkd1*, 5′-TGTCTGAAATCCTC GCAGTG-3′ and 5′-TCTCAAAGCAGTGTG-GGTTG-3′ to *Prkd2*, 5′-GGAGGC ATTGATTCTTCCTG-3′ and 5′-AAAGAGGCATCCACCACAAG-3′ to *Prkd3*, 5′-ATTAACGCGCAGATCATGCA-3′ and 5′-CCTCGGTGACGTTCAGATTGT-3′ to *Sod2*, 5′-AGGCACTGGAACTCGGCAATG-3′ and 5′-AAGGGCCCGAACAT ACGATT-3′ to *Bdnf*. Data were normalized to rat *Gapdh* and mRNA abundance was calculated using the $\Delta\Delta C_T$ method.

**Generation of PKD1 mutant and cloning in a lentiviral vector for neurospecific expression.** Mutant PKD1-Y[93]E/Y[469]E was generated by overlap PCR using pBSK-PKD1 mouse cDNA as a template as previously described[57]. Briefly, a sequence upstream of PKD1 cDNA corresponding to pBSK and close to the polylinker and a sequence near to the *Sph*I site within PKD1 were used as external forward (5′-GACTCACTATAGGGCGAATTGGGTACCG-3′) and reverse (5′-CTTGGGGATGACGGGCATAAGAGC-3′) primers together with internal complementary forward (5′-TCCCCGAATGTGGTTTC**GAG**GGACTCTATGATAAG ATC-3′) and reverse (5′-GATCTTATCATAGAGTCC**CTC**GAAACCACATTCGG GGA-3′) primers containing the residue substitution for Y[93]E (bold). After the second PCR reaction, the amplified fragment was digested with *Xho*I and *Sph*I, and the wild-type PKD-*Xho*I/*Sph*I fragment was then replaced. For double mutant PKD1-Y[93]E/Y[469]E, we followed the same approach, using as template pBSK-PKD1-Y[93]E mutant and internal primers forward (5′-ACAGGGAGCCGGTA C**GAG**AAGGAAATTCCTTTATCAGAA-3′) and reverse (5′-TTCTGATAAAGG AATTTCCTT**CTC**GTACCGGCTCCCTGT-3′) containing Y[469]E substitution (bold). The quadruple mutant PKD1-Y[93]E/Y[469]E/S[744]E/S[748]E was obtained by digesting pBSK-PKD1-Y[93]E/Y[469]E with *Xho*I and *Sph*I, and using this fragment to substitute that of pBSK-PKD1-S[744]E/S[748]E (a mutant obtained before)[57]. This quadruple mutant was digested with *Eco*RI and subcloned in p-EF-Bos-GFP as we have previously done[58]. Then, FL fusion sequence GFP-PKD1-Y[93]E/Y[469]E/S[744]E/ S[748]E was PCR amplified using forward (5′-GG**CCTAGGT**CACCATGGTGAGC AAGGGCGAGGA-3′) and reverse (5′-GG**TTAATTAAT**CAGAGGATGCTGAC ACGCTCACTG-3′) primers containing *Avr*II and *Pac*I sites, respectively (underlined). Once digested, the PCR product was cloned in a lentiviral vector bearing human Synapsin promoter (SYNpr) SYNpr-DsRed-SYN-GFP[59], where DsRed had been depleted after digestion with *Bam*HI and *Not*I and substituted by an adaptor with multiple cloning sites containing *Avr*II and *Pac*I sites (5′-GATCCC**CCTAGGC** GCGTTAATTAAGTTTAAACCTCGAGGC-3′ and 5′-GG**CCGCCTC**GAGGTTT AAAC**TTAATTAAC**GCGC**CCTAGG**G-3′; underlined). The second SYNpr and the GFP cassette were also eliminated by digestion with *Sph*I and *Eco*RI, klenow refilling and religation. Constructs were sequenced using an Applied Biosystems automated DNA sequencer (ThermoFisher Scientific).

**In vitro kinase assays.** HEK-293T cells were transfected with p-EF-Bos-GFP empty or containing wild-type PKD1 (GFP-PKD1), a kinase-dead mutant (GFP-PKD1-KD) or the quadruple mutant generated here with constitutive kinase activity (GFP-PKD1-Ca) for 48 h. Total lysates were immmunoprecipitated with anti-GFP antibodies and PKD1 kinase activity was assessed by in vitro kinase assay as described previously[57]. Briefly, immunoprecipitates were washed four times with RIPA buffer and twice with kinase buffer (30 mM Tris-HCl, pH 7.6, 10 mM $MgCl_2$, 2 mM dithiothreitol) containing [γ-$^{32}$P]ATP and incubated for 10 min at 30 °C. The reaction was stopped by adding sample buffer and immunoprecipitates were resolved by SDS–PAGE and transferred to nitrocellulose membranes. Filters were exposed to obtain autoradiography images and then subjected to immunoblot analysis with anti-GFP antibody.

**Plasmids for shRNA.** Lentiviral vectors containing shRNAs to interfere *Dusp1* (shDUSP1) or *Step* (shSTEP) expression were generated by cloning the following oligonucleotides into the *Hpa*I and *Xho*I sites of pLentiLox3.7 (pLL3.7): shDUSP1 5′-GTAACCACTTT GAGGGTCACTAGAATTCTAGTGACCCTCAAAGTGG TTATTTTTC-3′ and 5′-TCGAGAAAAATAACCACTTTGAGGGTCACTAGAA TTCTAGTGACCCTCAAAGTGGTTAC-3′, shSTEP 5′-GGCAGGCGAATTCTT TGAAATGAATTCATTTCAAAGAATTCCGCCTGCTTTTTC-3′, and 5′-TCGA GAAAAAGCAGGCGGAATTC TTTGAAATGAATTCATTTCAAAGAATTCC GCCTGCC-3′. Control shRNA vector (shC) was constructed by introducing oligonucleotides that do not match any known rat transcript: 5′-TCAACAAGATGA AGAGCACCAATTCAAGAGATTGGTGCTCTTCATCTTGTTGTTTTTTC-3′ and 5′-TCGAGAAAAAACAACA AGATGAAGAGCACCAATCTCTTGAATT GGTGCTCTTCATCTTGTTGA-3′. The targeted *Dusp1* sequence corresponds to

**Fig. 7** In vivo neuroprotection by PKD1-Ca from KA-induced hippocampal neuronal death. **a** Adult male Wistar rats were stereotaxically grafted in the hippocampal CA1 region with GFP (right hemisphere) or PKD-Ca (left hemisphere) lentivirus. (Right panel) Scheme of the KA injection protocol used. After 12 days of lentiviral grafting, rats were treated with saline or KA (8.5 mg per kg, i.p.) and killed 3 days later. **b, c** Two consecutive brain series were double stained with NF-κB or NeuN and anti-GFP antibodies to identify transduced neurons in CA1. **b** Representative confocal images and zoom-boxed regions are shown from saline treated animals (n = 6 sections per animal, n = 3 animals). **c** Representative confocal images from right and left CA1 regions in the same slice from an animal treated with KA (n = 6 sections per animal, n = 4 animals). See zoom-boxed regions for details of nuclear staining of NeuN and NF-κB, showing it is more preserved in PKD1-Ca-transduced neurons (right panels) compared to the contralateral side GFP-transduced neurons (left panels). **d** Representative FluoroJade B staining from brain slices showing neuronal death at the CA1 region from saline (n = 3) and KA-treated (n = 4) animals. (Right panel) Quantification of neuronal death as the percentage of FluoroJade B signal in the right or left hemisphere in KA-treated animals (n = 6 sections per animal, n = 4 animals). Data are expressed as mean + s.e.m. analyzed with two-tailed unpaired Student's *t* test. **P < 0.01. **e** Scheme illustrating the role of PKD in healthy or degenerating neurons. Healthy neurons present basal moderate levels of active PKD and IKK. In this condition, IKK controls IκB phosphorylation and proteasomal degradation, and NF-κB localizes at the nucleus. Nuclear NF-κB activates *Sod2* transcription, increasing SOD2 levels to facilitate ROS elimination. In excitotoxic neurodegeneration, high concentrations of glutamate overstimulate NMDARs and KARs, and drive PKD dephosphorylation and inactivation through the induction of PP1 and DUSP1 phosphatases. This step contributes to the shut-off of the IKK/NF-κB/SOD2 axis, decreasing SOD2 levels, and promoting ROS accumulation, oxidative damage and neuronal death

rat mRNA positions 782–804 and Step sequence corresponds to rat mRNA positions 1320–1342. The integrity of these novel constructs was verified by sequencing. The two Prkd1 shRNA lentiviral vectors (a and b) and the one for Prkd2 were obtained from Sigma-Aldrich.

**Lentiviral production and transduction of neuronal cultures.** Lentiviral suspensions were prepared in HEK293T cells as previously described[23]. Briefly, HEK293T were transfected with lentiviral vectors and packaging vectors using Lipofectamine 2000 reagent and OPTI-MEM media (ThermoFisher Scientific Gibco) for 4 h following the manufacturer instructions. Medium was then changed to IMDM complemented with 5% fetal bovine serum, 5% horse serum, 22.2 mM glucose, 0.1 mM Glutamax-I (ThermoFisher Scientific Gibco), penicillin (100 U per ml), and streptomycin (100 U per ml). Supernatant was collected after 48 h. For concentration, the viral suspension was first filtered using a Steriflip-HV 0.45-μm filter unit (Millipore, Billerica, MA, USA) and ultracentrifuged at 20,000 rpm for 2 h at 4 °C in a SW28 Beckman Coulter rotor. Viral pellets were resuspended overnight at 4 °C in PBS. HEK-293T cell line was purchased from ATCC and tested to be mycoplasma-negative. DIV7 neurons were transduced with concentrated lentiviral suspensions ($10^7$–$10^8$ pfu per ml) directly added to the growing media for 7 additional days.

**Assessment of neuronal viability in neuronal cultures.** Analysis of neuronal survival was measured by MTT reduction assay (Sigma-Aldrich), as previously described[23]. Briefly, MTT (0.5 mg per ml) was added to the medium of neuronal cultures, and after 2 h at 37 °C, medium was eliminated and the formazan salts formed were solubilized in 100 μl of DMSO and spectophotometrically quantified at 570 nm. Data were represented as the percentage of neuronal viability giving to the control conditions a 100% value. Neuronal death was also analyzed by determining nuclear condensation and neuronal shape, visualized by DAPI and MAP2 staining, respectively, and expressed as percentage of nuclear condensation. Viability of neurons transduced with GFP or PKD1-Ca lentiviral particles was expressed as the percentage of GFP+ neurons after NMDA or KA treatments relative to control untreated neurons taken as 100% viability.

**Determination of ROS production in cultured neurons.** Rat cortical neurons DIV14 were treated with NMDA for 1 h and then washed and loaded with 10 μM Cell ROX Deep Red Reagent fluorogenic probe (ThermoFisher Scientific), by incubating them at 37 °C for 30 min in the dark. The medium was removed and cells were washed once with PBS, trypsinised and collected. Then, cells were washed twice with PBS and fixed with 4% PFA for 10 min at room temperature. Cellular fluorescence intensity was measured on a FACSCanto II flow cytometer (BD Biosciences, Allschwil, Switzerland). For each analysis, 10,000 events were recorded. ROS production was estimated using the mean fluorescence intensity of each cell population. In neurons transduced with GFP or PKD1-Ca, the mean of the fluorescence intensity relative to the total of GFP+ population was measured.

Cultures of mouse cortical neurons from PKD1[floxed] and PKD1-KO mice were treated with NMDA for 1 h and then washed and loaded with 10 μM dihydroethidium (DHE, ThermoFisher Scientific), by incubating them at 37 °C for 30 min in the dark. The medium was removed and cells were washed once with PBS, trypsinised, and collected. Then, cells were washed twice with PBS and fluorescence intensity was measured by flow cytometry as above. For each analysis, 10,000 events were recorded. ROS production was estimated using the mean fluorescence intensity of each cell population. In addition, DHE staining of neurons seeded onto coverslips was also analyzed by confocal microscopy.

**Luciferase assays.** Cortical neurons plated in 24-well plates were transfected at DIV7 with (NF-κB)3-tk-Luc or tk-Luc, together with renilla luciferase plasmids. Cells were transfected in serum-free medium by using 0.5 μg of DNA and 0.5 μl of Lipofectamine 2000 reagent (Invitrogen) per well, according to the manufacturer's specifications. Two days after transfection, neurons were treated with NMDA for 3 h and processed for luciferase activity measurements. Luciferase activity was detected by adding the enzyme substrate provided in the Dual-Glo™ luciferase assay system (Promega Corporation), according to the manufacturer's instructions. The samples were assayed on a Veritas microplate luminometer (Turner BioSystems Inc., California, USA).

**Ca²⁺ measurements.** Single-cell $Ca^{2+}$ levels in cortical neurons transduced with GFP or GFP-PKD1-Ca lentiviral particles were recorded using the ratiometric $Ca^{2+}$ indicator dye Fura Red acetoxymethyl ester (ThermoFisher Scientific). Cells were grown on eight-well chamber slides and loaded with 5 μM Fura Red, AM for 30 min at 37 °C in HBSS containing 145 mM NaCl, 5 mM KCl, 0.75 mM $Na_2HPO^4$, 10 mM glucose, 10 mM HEPES (pH 7.4), 1 mM $MgCl_2$, and 2 mM $CaCl_2$ (high $Ca^{2+}$ medium), and then rinsed and left undisturbed for 30 min at 37 °C to allow for de-esterification. Measurements of intracellular $Ca^{2+}$ levels were performed every second at 37 °C using a Confocal LSM510 META microscope (Zeiss, Germany). Changes in intracellular $Ca^{2+}$ concentration ($[Ca^{2+}]i$) in individual neuronal cell bodies are expressed as the F458/F488 ratio after subtracting background fluorescence. This ratio represents the emission intensities at 660 nm obtained after excitation at 458 and 488 nm. Cells that responded rapidly to NMDA stimulation

were identified as neurons and only GFP+ neurons were analyzed. Data processing was performed using ImageJ software.

**Immunofluorescence of cultured neurons.** Primary neurons grown on coverslips were fixed with 4% PFA with 4% sucrose, permeabilised with 0.1% Triton X-100 in PBS for 5 min and then incubated in 4% bovine serum albumin in PBS for 30 min. Coverslips were then incubated for 1 h at room temperature with the appropriate primary antibody in blocking solution followed by the corresponding secondary antibody. The nuclei were stained with DAPI and coverslips were mounted with Prolong medium (ThermoFisher Scientific).

**Immunohistochemistry of rat and mouse and human brain samples.** Rats and mice were anesthetized by an i.p. injection of pentobarbital and then perfused with 4% PFA. Brains were removed and post-fixed in 4% PFA overnight and cryoprotected in 30% sucrose in PBS for 48 h before freezing. Thirty μm-thick coronal sections were obtained by a Cryostat (Leica Microsystems, Germany). Floating coronal sections were incubated in blocking solution (1% BSA, 1% Triton X-100 in 0.1 M PBS) with primary antibodies overnight at 4 °C and subsequently incubated 2 h at 4 °C with secondary antibodies. Then, slices were rinsed in PBS and nuclei were stained with DAPI and mounted with Prolong medium.

In a complete series of coronal brain sections from KA-treated rats injected with lentivirus, transduced neurons were identified in the CA1 hippocampal region by immunostaining with anti-GFP antibodies. The same series of sections was also co-stained for NeuN and NF-κB. All sections containing GFP signal were analyzed by confocal microscopy imaging for NeuN and NF-κB. An additional series of sections was mounted on slides for Nissl staining with cresyl violet to see KA-induced neuronal death at the CA1 hippocampal area.

Complete series of coronal sections from PKD1[floxed] and PKD1-KO mice were stained free-floating using the biotin–avidin peroxidase method and 3-3′ diaminobenzidine (DAB; Sigma FAST DAB, Sigma-Aldrich) as a chromogen. Selected pictures correspond to coronal slices from the rostral region of brain cortex (Bregma +1.34 mm to −0.84 mm). Endogenous peroxidase was inactivated by incubating sections in a solution of 0.3% hydrogen peroxide in PBS for 30 min. Brain sections were pretreated for 1 h with 1% BSA, 5% FBS, and 0.2% Triton X-100 in PBS, and subsequently incubated with rabbit polyclonal antibodies against PKD and phospho-PKD-S[916]. Finally, brain sections were incubated with avidin–biotin complex using Elite Vectastain kit (Vector Laboratories, Burligame, CA, USA). Chromogen reactions were performed with DAB and 0.003% hydrogen peroxide for 10 min before mounting with Prolong medium.

Complete series of coronal sections from Sham- and MCAO-operated wild-type mice or PKD1[floxed] and PKD1-KO animals were Nissl stained to identify slices that contained areas of ischemic damage and corresponded to ischemic areas initially identified by T2W MRI. Additional series of slices were also stained for the analysis of the different markers (NeuN, and p-PKD or NF-κB or MDA) by confocal microscopy. To assess the histological sections in a stereological fashion, the anatomical extent of neuronal injury in the ischemic hemisphere was defined based on the NeuN staining, showing damage in infarcted areas. The regions containing neuronal injury were not well captured by standard anatomical definitions, so specific regions of interest (ROI) boundaries were determined in the cortex adjacent to the nucleus of the infarct for each coronal slice. Specifically, the ROI was defined by blinded observers in the rostral region containing the striatum and primary somatosensorial cortex and primary motor cortex (Bregma +1.34 mm to −0.84 mm)[60] and whose boundaries were the external capsule, the secondary motor cortex and the granular insular cortex. The ischemic core localized extensively in the striatal area, where no NeuN, and p-PKD or NF-κB signal was detected, hindering any quantitative analysis of this region. Quantification studies were performed in the penumbra area of three independent selected sections per animal after defining ROI boundaries as explained above.

Paraffin-embedded human postmortem stroke 15-μm sections were processed, treated with citrate buffer (pH 6.0) for antigen retrieval, and incubated in 10% donkey serum for 1 h. For double immunofluorescence analysis, sections were incubated with primary antibodies for 2 days in blocking solution followed by washes in PBS and incubation with secondary antibodies in the same solution. Then, slices were rinsed in PBS and nuclei were stained with DAPI. Sections were treated with a saturated solution of Sudan black B (Merck Bioscience, Darmstadt, Germany) for 15 min to block autofluorescence of lipofuscin granules. The sections were later washed and mounted with Prolong medium.

**FluoroJade B staining of rat brain.** To quantify brain damage in KA-treated rats, a different set of coronal brain sections adjacent to that used for immunofluorescence analysis was stained for in situ cell death using FluoroJade B staining (Millipore Corporation, Billerica, MA, USA). It is important to point out that we confirmed that GFP fluorescence was not affecting FluoroJade B staining. For that purpose, we combined FluoroJade B labeling with GFP immunostaining using an anti-GFP antibody detected with a secondary Alexa-555 antibody and found absolutely no co-localization of both signals by confocal microscopy analysis. For FluoroJade B staining, sections were mounted in superfrost slides and incubated for 3 min with absolute ethanol followed by 1 min in 70% ethanol. After rinsing in $H_2O$, slides were immersed in 0.06% $KMnO_4$ for 15 min in the dark. Then, slides

were extensively washed in $H_2O$ and transferred to a staining solution containing 0.1% acetic acid and 0.0004% FluoroJade B for 30 min in the dark. Slides were again rinsed in $H_2O$, dried, and submerged directly into xylene and mounted in DPX medium (Sigma-Aldrich). For quantification analysis, CA1 regions were outlined, and total FluoroJade B staining in the outlined region was measured using a densitometric thresholding technique implemented with ImageJ 1.47d software (NIH). The threshold was set at a level just above that with counted background and non-specific staining in areas outside the outlined region. The analysis was done in a series of sections through the entire hippocampus from each rat brain. Data were represented as the % FluoroJade B staining and indicate the total number of stained pixels over threshold.

**Brain vascular anatomy**. PKD1$^{floxed}$ and PKD1-KO mice littermates brain vascular anatomy was analyzed as described[61]. Briefly, mice were anesthetized by an i. p. injection of pentobarbital and then 2 ml of 5% Evans blue solution in saline was perfused intracardially for 1 min. Animals were killed and brains were fixed by immersing the skulls in 4% PFA for 48 h. Macroscopic images of ventral and dorsal views of the brain were acquired under a stereo microscope (M50, Leica, Germany) under cold light illumination using a digital camera (Infinity 2 Digital Camera, Nikon). Quantification of dorsal surface brain images was performed with ImageJ software, measuring the distance between the anastomotic line, drawn as previously described[62], and the midline at 3 and 6 mm from the frontal pole of the brain.

**Images acquisition**. Confocal microscopy images were acquired using plan-apochromatic objectives in an inverted Zeiss LSM 710 laser scanning microscope (Zeiss, Germany). To avoid crosstalk between channels, sequential scanning mode was used. All images shown correspond to the maximum intensity projection of serial sections; in case of different projections, details can be found in figure legends. Pictures were processed with Zen 2009 (Carl Zeiss MicroImaging), Adobe Photoshop CS (Adobe Systems Inc.), and ImageJ 1.47d (NIH) software. Nissl and DAB staining bright field images were captured in a Nikon Eclipse 90i microscope, using a Digital Sight DS-QiMc camera and NIS-Elements BR 3.0 software.

**Quantitative and statistical analysis**. Immunoblot signals were quantified by densitometric analysis (NIH Image) and normalized using NSE. Phospho-antibodies signals were normalized using the total protein values relative to those of NSE. Data were expressed relative to values obtained in their respective untreated controls. Neuronal viability in cortical cultures was measured by MTT or DAPI condensation. Three independent experiments were quantified in MTT assays. The mean of these experiments was represented as the % of neuronal viability giving a 100% value to the control conditions. DAPI condensation was quantified counting 120–250 cells and expressed as the % of nuclear condensation relative to the total number of cells for each condition. Neuronal viability in cortical neurons transduced with GFP or PKD1-Ca was performed counting the number of GFP+ cells relative to the total number of neurons present in control conditions taken as 100% viability. Nuclear localization of NF-κB in neuronal cultures was analyzed by manual counting the presence or absence of the protein in the nuclei of 50–100 neurons (MAP2+ cells) per sample relative to control untreated cells. In neurons transduced with GFP or PKD1-Ca, we counted the number of cells with nuclear NF-κB in 50–100 GFP+ cells and represented it relative to the total number of cells for each condition. For the analysis of immunohistochemistry in MCAO-operated mice, the intensity of p-PKD-S916 and NF-κB staining was measured in 100 NeuN+ neurons and compared to the value of the mean obtained from sham-operated animals. In human samples, the presence or absence of p-PKD-S916 and NF-κB staining was analyzed by manual counting in 30–50 NeuN+ neurons for each condition (control and ischemic stroke). Analyses of significant differences between means were carried out using unpaired or paired two-tailed Student's $t$ test with Welch's correction when appropriated. No statistical method was used to predetermine sample size. Sample sizes were based on previously published experiments by us and others, and on our own unpublished data. The experiments were not randomized. The investigators were not blinded to allocation during experiments or outcome assessment. Results are shown as mean ± s.e.m. and the number of experiments carried out with independent primary neuronal cultures and animals ($n$) is shown in figure legends.

**Data availability**. The data sets generated and/or analyzed during the current study are available from the corresponding author on reasonable request.

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

## Acknowledgements

This work was supported by grants SAF2014-52737-P to T.I., SAF2013-45258-P to M.R.C., BFU2016-77885-P to F.H., SAF2014-54070-JIN to A.M. from Ministerio de Economía, Industria y Competitividad (Spain). It was also funded by P2010/BMD-2332 (Neurodegmodels) from Comunidad de Madrid to T.I., F.H. and J.A.) and Centro de Investigación Biomédica en Red sobre Enfermedades Neurodegenerativas (CIBERNED, Instituto de Salud Carlos III, Spain) to T.I., F.H., J.A., and I.F.). J.P.-U. is a recipient of a predoctoral contract from SAF2014-52737-P; L.G.-G. was funded by a contract from CIBERNED; A.G.-M. and A.S.S. were funded by contracts from CIBERNED cooperative projects 2013/07 and CIBERNED 2015-2/06, respectively. A.M., A.D.P. is a recipient of a Juan de la Cierva formación fellowship (Ministerio de Economía, Industria y Competitividad, Spain) associated to CIBERNED. J.J.-A. was funded by a predoctoral contract from CSIC (JAEPredoc program) and by CIBERNED. The cost of this publication has been paid in part by FEDER (European Funds for Regional Development) funds. We are grateful to Professor Eric N. Olson (Department of Molecular Biology, University of Texas Southwestern Medical Center, Dallas, USA) for kindly providing *Prkd1*-floxed mice; Professor Jianping Ye (Pennington Biomedical Research Center, Baton Rouge, LA, USA) for his generous gift of RelA−/− mouse embryo fibroblasts; Dr. Marta Nieto (Centro Nacional de Biotecnología, Madrid, Spain) for providing reagents; Dr. C. López-Menéndez for scientific advice an critical reading of this manuscript as well as members of our laboratories for constructive suggestions. We also acknowledge Dr. L. Sánchez-Ruiloba for her valuable support with confocal microscopy, Drs. M. Díaz-Guerra and S. Gascón for providing lentivirus containing synapsin promoter and helpful discussion, Drs. Ana Guadaño-Ferraz and S. Bárez-López for valuable advice in human and mouse brain immunohistochemical approaches. We also thank M. Rodríguez-Martínez and T. Navarro for performing MRI analysis; J. Pérez for image design; and I. Santana, S. Ramirez, S. Valle, G. Cano-García, J. Deyell, and J. Henández-Bermúdez for their help and technical support.

## Author contributions

T.I., M.R.C., M.L., A.M., L.K., L.G.-G., and J.P.-U. conceived and designed the experiments. L.G.-G., J.P.-U., A.D.P., A.M., J.J.-A., N.S.De.L.-R., A.G.-M., A.S.S., M.G.-G., M.J.P.-A., and I.F. performed experiments, analyzed, and discussed results. J.F., C.I., F.H., J.A. contributed reagents, animals, materials or analysis tools, and discussed results. T.I., M.R.C., N.S.De.L.-R., L.G.-G., and J.P.-U. wrote the manuscript with feedback from all authors.

## Additional information

**Competing interests:** The authors declare no competing financial interests.

