## [Peer Review File · Nature Communications]

Reviewers' comments:

Reviewer #1 (Remarks to the Author):

The mechanisms by which excitotoxicity-induced oxidative stress induces neurodegeneration are largely unknown. In this manuscript the authors show that a constitutive neuronal PKD1/NF-kappaB pathway is important for detoxification from oxidative stress. Moreover, PKD1 can rescue the excitotoxic shut off of this pathway.

Overall, these are interesting findings putting a previously described ROS-PKD1-IKK-NFkappaB signaling pathway into new physiological context. The author also identified two phosphatases which regulate PKD1 – a new important finding with respect on how PKD is regulated.

Although I think that this manuscript once revised can be an important contribution to the field, I have two specific issues with the manuscript in its current form: One is that it is unclear which isoform of PKD the authors detect in their Western blotting experiments. Especially, data obtained with phospho-specific antibodies which cross-react with other isoforms should be validated by analyzing phosphorylation after immunoprecipitation of PKD isoforms (see my comments to Figure 1). This is important since it becomes increasingly important to distinguish between redundant and specific functions of these enzymes. Another issue is that although the authors put everything in context to Src- and tyrosine phosphorylation-regulated activation of PKD1, they never show this (i.e. by probing for these tyrosine phosphorylations).

Figure 1: in Figure 1b, the point is made that PKD is basally active and can be further activated by NMDA. However, this does not become clear with the blots shown. The blots that were included show almost no staining with the pS916 or the pS744/748 antibodies. Why is S744/748 phosphorylation not increasing after 5 min NMDA treatment? It is also unclear which of the three PKD isoforms are activated. Due to the increase in S916 phosphorylation one can assume that is PKD1 or PKD2. It is important that the authors define the expression pattern of PKD1, PKD2 and PKD3 (i.e. by qRT-PCR) in cells used, and then defined which of the isoform is responsive to NMDA. The authors also should comment on why the blots for PKD show a single and the pS916 and pS744/748 show doublets (are other PKD isoforms picked up?). Is the PKD antibody used specific for PKD1? Generally, the antibodies used are not defined in the Materials and Methods section. This makes it impossible to judge which isoform is detected. It is imminent that information is added that allows reproducibility (i.e. catalogue number) for each antibody used. This can be included as a table. In Figure 1f, the inhibitors used are quite non-specific. A knockdown approach or adding another set of inhibitors (i.e. SU6656 and an nPKC inhibitor) may be another option. In addition to this the authors should include staining for Src induced phosphorylations at Y93 and Y469.

Figure 3: In Figure 3c and 3d, in order to exclude off-target effects, it is standard to either use as second shRNA, or to perform a rescue experiment (i.e. using a PKD1 from a different species). One of these needs to be included. Figures 3f to 3i show only controls and should

go to the Supplemental Data. In Figure 3j, the tissues obtained with the PKD1 k/o or control mice should also be stained by immunohistochemistry for pS910 and PKD1. These are important controls. In addition, IHC for phosphorylation of oxidative stress-mediated tyrosine phosphorylations of PKD1, as well as markers for oxidative stress (i.e. 4-HNE) should be included.

Do antioxidants have the same effects that the PKD1 inhibitor in cells or in vivo?

In the introduction (page 4) the author's state: "Oxidative stress is an important activator of PKD1 in cellular models, but its capacity to activate this kinase in vivo has not been confirmed". This is actually not correct. PMID 26947075 have shown that the ROS-PKD-NF-kappaB pathway is a driver of pancreatic cancer development. On page 5, a similar incorrect statement is made.

The quality of the immunofluorescence shown in Figure 2d needs to be improved.

Molecular weight markers need to be added to the Western blots.

Reviewer #2 (Remarks to the Author):

In this study García-Guerra et al investigate the role of PKD1 in excitotoxic neuronal injury and brain ischemia. The authors show that PKD1 is transiently activated after excitotoxic glutamate receptor activation in vitro and in two excitotoxic in vivo models, transient focal ischemia in mice and kainic acid neurotoxicity in rats. The authors hypothesize that post-activational down-regulation of PKD1 activity contributes to excitotoxic neuronal cell death. Using a comprehensive pathway analysis approach the authors show that loss of PKD1 activity possibly due to dephosphorylation by DUSP1 results in deactivation of the enzyme after excitotoxic NMDA challenge. PKD1 auto-phosphorylation was associated with increased NF-kappaB activity, reduced ROS production, and neuronal survival. Although the study uses a variety of assays to address the molecular components involved in PKD1-mediated cell survival, the assays are not applied consistently to all paradigms. Specific concerns are:

1. The majority of studies, including those from the Schwaninger laboratory, show that excitotoxicity and ischemia induces neuronal NF-kappaB activation. These findings are in contrast to data presented in the current study. This leads to two essential questions: 1) Is the antibody to p65/RelA specific? This could be tested in RelA^{-/-} cells. 2) Is nuclear translocation of RelA indeed reflective of NF-kappaB activity. While nuclear translocation is clearly a prerequisite for NF-kappaB mediated transcription, it is not always sufficient. Determining the S536 phosphorylation state might give further insight of NF-kappaB transcriptional activity. In addition, EMSA assays of nuclear extracts might be useful to detect increased NF-kappaB DNA binding activity.
2. The mechanism of NF-kappaB induction by PKD1 remains elusive. What are the molecular targets of PKD1? Is it acting directly on IKK or other upstream kinases involved in NF-kappaB activation?

3. ROS production should also be tested in PKD1 KO neurons. In addition, in vivo data are needed to determine whether the regulation of ROS generation found in cultured neurons are also observed in the ischemic and kainite model in vivo. In addition, the 24 hours endpoint seems rather late to assess pathways involved in cell death in the ischemic model because excitotoxic neuronal loss might already been maximal at this time.
4. Fig 1d: It is not clear why AP5 treatment increases DPK1 activity at baseline. Does synaptic activity inhibit PKD1?
5. Activity and activation are used synonymously. Kinase phosphorylation at one residue is not necessarily predictive of its activity.
6. The activity of the constitutive active DPK1 mutants should be tested in an in vitro kinase assay.
7. Cellular experiments using phosphatase inhibitors are more complex than stated by the authors. Increased phosphorylation of DPK1 upon inhibitor treatment might not indicate that these changes are due to direct activity of the respective phosphatases on the target. Most likely, some of these inhibitors are also resulting in increased activity of upstream protein kinases which could potentially result in increased PKD1 phosphorylation.
8. In some vitro experiments the dosage of NMDA is not evident (e.g figure 1b, 5c).
9. IKK1 and IKK2 might have opposing effects on NF-kappaB activity. It is not clear which of the two enzymes has been investigated in this study. Furthermore, the NF-kappaB activation should be tested on the transcriptional level by monitoring the mRNA expression levels of selected NF-kappaB-dependent genes including SDO2 in vitro and in vivo.
10. Were DPK1^{flox/flox} and DPK1 KO mice derived from littermates? What is the genetic background of these animals. The authors should also provide an assessment of macroscopic brain vascular anatomy of DPK1 KO mice.
11. The assessment of histological sections should have been performed in a stereological fashion. It is not clear how representative the current analysis is.

Reviewer #3 (Remarks to the Author):

The authors' findings are interesting and novel. However, there are several concerns that reduce enthusiasm for the manuscript.

1.. The authors are advised not to use the term "excitotoxicity-induced oxidative stress" as it implies that oxidative stress is the key mediator of excitotoxic neuronal death. Excitotoxicity can manifest as necrosis which results from rapid massive Na⁺ influx and cell swelling and membrane rupture, or as apoptosis triggered by Ca²⁺ influx and uptake into mitochondria / PTP opening / cytochrome c release / caspase 3 activation... While oxidative stress certainly occurs in excitotoxic neuronal death, it is not a pivotal early event. Also, while the authors focus on genes encoding antioxidant enzymes as NF-kB targets in neurons (which is true), NF-kB also induces genes encoding anti-apoptotic Bcl-2 family members and Ca²⁺-regulating proteins that can protect against excitotoxicity.

2. PP1 is activated by Ca²⁺ and it is likely Ca²⁺ influx that causes dephosphorylation of PKD1.

3. Figure 2a and b. Most of the striatal neurons in the region examined are killed by the ischemic insult. It is therefore not surprising that little or no p-PKD is seen. Animals should be killed at earlier time points 2 – 8 hours after the ischemic insult, before the neurons are dead.

4. It is clear from the images of cultured neurons shown in Figures 3 and 4, and from an extensive literature of published studies of excitotoxicity in primary neuronal cultures, that with the concentration of NMDA used the neurons are dying of rapid (within 1 – 2 hours) necrosis (cell swelling and rapid neurite beading and fragmentation). Excitotoxic necrosis cannot be prevented with caspase inhibitors, and I expect that caspase inhibitors will not protect the neurons in the present study. When neurons die by excitotoxic apoptosis, the cell death is delayed for 12 – 24 hours, and the cells shrink in size (this death can be inhibited with caspase inhibitors).

5. When neurons die by excitotoxic necrosis their nuclei often exhibit non-specific immunoreactivity with many different antibodies. This may be the case in the present study, resulting in nuclear staining with the NF- κ B subunit antibody.

6. The excitoprotective effect of the dephosphorylation-resistant PKD1 mutant is impressive. This may be due to an effect of p-PKD1 on Ca^{2+} influx or buffering. Ca^{2+} imaging studies are required to rule this in or out.

Reviewers' comments (black) and **Authors' answers (blue)**:

NOTE: Main manuscript text, figure legends and Supplementary information text and legends contain information relative to new experiments, results and modifications labelled in “blue”.

Reviewer #1 (Remarks to the Author):

The mechanisms by which excitotoxicity-induced oxidative stress induces neurodegeneration are largely unknown. In this manuscript the authors show that a constitutive neuronal PKD1/NF-kappaB pathway is important for detoxification from oxidative stress. Moreover, PKD1 can rescue the excitotoxic shut off of this pathway. Overall, these are interesting findings putting a previously described ROS-PKD1-IKK-NFkappaB signaling pathway into new physiological context. The author also identified two phosphatases which regulate PKD1 – a new important finding with respect on how PKD is regulated.

Although I think that this manuscript once revised can be an important contribution to the field, I have two specific issues with the manuscript in its current form: One is that it is unclear which isoform of PKD the authors detect in their Western blotting experiments. Especially, data obtained with phospho-specific antibodies which cross-react with other isoforms should be validated by analysing phosphorylation after immunoprecipitation of PKD isoforms (see my comments to Figure 1). This is important since it becomes increasingly important to distinguish between redundant and specific functions of these enzymes. Another issue is that although the authors put everything in context to Src- and tyrosine phosphorylation-regulated activation of PKD1, they never show this (i.e. by probing for these tyrosine phosphorylations).

We thank the Reviewer for his/her comments on the interest and importance of the study, and also for raising important issues that we have taken into account to improve the manuscript. We have performed further experiments that support and validate our hypothesis and address the Reviewer's concerns.

COMMENT 1

Figure 1:

1) in Figure 1b, the point is made that PKD is basally active and can be further activated by NMDA. However, this does not become clear with the blots shown. The blots that were included show almost no staining with the pS916 or the pS744/748 antibodies. Why is S744/748 phosphorylation not increasing after 5 min NMDA treatment?

We agree that the signal of the bands detected with pS916 or pS744/748 antibodies is sometimes faint and have improved the corresponding images as much as we could in the revised version of the manuscript (i.e. **Figure 1b**). However, it should be considered that we are detecting endogenous PKD phosphorylation in primary neurons cultured for 14 days. These blots are not coming from overexpressing PKD or from cell lines lysates. Neuronal cultures are heterogeneous and signal intensity or basal PKD activation can vary slightly in each culture. Despite of this, we can detect a marked and significant increase in the signal under different activatory conditions and its disappearance shortly after exposing neurons to excitotoxicity. Moreover, we can potentiate that signal by using several pharmacological inhibitors, including phosphatase inhibitors. These changes are reproducible (as shown through other panels in **Figure 1** and in many others) and statistically significant, as shown in the graph on the right (**Fig1b**) for pS916 antibody signal quantification. In addition, we have quantified now pS744/748 signal in immunoblots from n=3 experiments to show a very similar result to the one obtained with pS916 antibody, a significant increase after 5 min and a significant decrease at longer time points (see **New Supplementary Figure 1j and the text included in the manuscript: Page 8, line 177-180**). In summary, the conclusion is the same as we stated in our initial version of the manuscript: neurons show basal PKD activity, excitotoxic concentrations of NMDA produce an early activation (shown by Ser916 and S744/748 phosphorylation) followed by a sharp dephosphorylation.

2) It is also unclear which of the three PKD isoforms are activated. Due to the increase in S916 phosphorylation one can assume that is PKD1 or PKD2. It is important that the authors define the expression pattern of PKD1, PKD2 and PKD3 (i.e. by qRT-PCR) in cells used, and the authors also should comment on why the blots for PKD show a

single and the pS916 and pS744/748 show doublets (are other PKD isoforms picked up?). Is the PKD antibody used specific for PKD1?

The reviewer is correct when assuming that increases in pS916 may reflect increases in PKD1 or PKD2 activity because PKD3 is not having the same C-terminus (containing Ser916) as PKD1 and PKD2. To address this reviewer's concerns, we transduced neuronal cultures with lentivirus encoding control shRNA (shC), shPKD1, shPKD2, or shPKD1+shPKD2. Immunoblot analysis using pS916 (Cell Signaling Technology #2051) showed that shC-transduced neurons contained a doublet. The upper band of this doublet disappeared after silencing PKD1 while the lower band did it after downregulating PKD2. The doublet was not detectable after using the shPKD1+shPKD2 lentivirus combination. These results indicate that the upper band detected by pS916 corresponds to PKD1 while the lower band corresponds to PKD2. By contrast, analysis of the same lysates with total PKD antibody (Cell Signaling Technology #2052, against a the C-terminal epitope of PKD1 that is highly similar to that of PKD2, containing S916) suggests that it preferentially detects PKD1 over PKD2, since the single band detected by this antibody disappeared after silencing PKD1 but not PKD2. These results are shown in **New Supplementary Figure 1a** and the corresponding text has been included under **Results (Page 6 line 139-141)** and **under Methods (information relative to shPKD and shPKD2 lentivirus)**. We conclude that although the antibody used to detect PKD can potentially react with PKD1 and PKD2, it detects preferentially PKD1 in neuronal lysates.

Finally, to define the expression pattern of PKD1, PKD2 and PKD3, we have performed qRT-PCR, as suggested by the reviewer. We could see that PKD1 is the most abundant mRNA, expressed two folds over mRNA levels of PKD2 and PKD3, and that NMDA treatment did not have any effect on the mRNA content of any of the three isoforms. These results are shown in **New Supplementary Figure 1b-1c** and described under **Results (Page 6 line 141-144)**. Related **Methods** have been included also (**Methods: information relative to qRT-PCR and primers used for PKD1, 2, 3 specific amplification**).

3) Generally, the antibodies used are not defined in the Materials and Methods section. This makes it impossible to judge which isoform is detected. It is imminent that information is added that allows reproducibility (i.e. catalogue number) for each antibody used. This can be included as a table.

Following this reviewer's suggestion, we have included a **New Supplementary Table 2** with the information relative to the antibodies used in our study, introduced under **Methods (Commercial Antibodies)**.

4) In Figure 1f, the inhibitors used are quite non-specific. A knockdown approach or adding another set of inhibitors (i.e. SU6656 and an nPKC inhibitor) may be another option.

We thank the reviewer for this suggestion. We have used now Rottlerin (as PKC δ inhibitor) and SU6656 (as Src family inhibitor) to treat neuronal cultures and obtained similar results to those obtained with the more general inhibitors GFI and PP2. These results are shown as a **novel panel in Figure 1f, and their quantification in new Supplementary Figure 3b**. We have moved the results with GFI and PP2 to **new Supplementary Figure 3a**. Text regarding these sets of data has been included under **Results (Page 8-9 line 198-203)**.

5) In addition to this the authors should include staining for Src induced phosphorylations at Y93 and Y469.

Following this reviewer's suggestion, we have used polyclonal antibody (Abcam ab59415), recognizing pY463 in human PKD1, the equivalent residue to mouse pY469. We have also employed a novel unpublished mouse monoclonal antibody (mAb) that we have generated against human pY95 that recognizes mouse pY93. **Novel Supplementary Figure 2** and a corresponding **section in Methods (Generation of phosphospecific PKD1 p-Tyr93 monoclonal antibody, Page 23 line 538)** contain all the information relative to the generation, characterization and phosphospecificity of this antibody.

Western blot staining with pY463 antibody showed a faint band in lysates from neurons treated with the tyrosine phosphatase inhibitor pervanadate but not in untreated cells (**Supplementary Figure 2a**). Our novel pY93 mAb detected a band corresponding to PKD after pervanadate and H₂O₂ treatment but not in unstimulated cells (**Supplementary Figure 2a and 2c**). Importantly, Y93 phosphorylation was sensitive to the Src inhibitor SU6656 (**Supplementary Figure 2b**), indicating that this tyrosine kinase participates in PKD phosphorylation in neurons.

The pY93 mAb was also able to immunoprecipitate PKD phosphorylated at Y93 from neuronal lysates after pervanadate or H₂O₂ treatment, as assessed by immunoblot analysis with the same antibody (**Supplementary Figure 2c**). However, immunoprecipitates obtained from untreated neurons showed only a very faint band (**Supplementary Figure 2c**, see immunoprecipitates from untreated neurons in the panel of H₂O₂ treatment) that was not always visible in every experiment. Unfortunately, pY463 signal was not detected in pY93 immunoprecipitates (not shown). This set of results suggests that PKD Y93 and Y469 basal phosphorylation in neurons may be very dynamic, having phosphatases tightly controlling them in continuous on/off cycles. This fact would therefore difficult their detection unless phosphatase inhibitors (such as pervanadate) or strong activators (such as high concentrations of H₂O₂) were used.

As a general comment, tyrosine phosphorylation could be weaker or more fragile or very sensitive to tyr-phosphatases actions than that for serines and/or the phospho-specific tyrosine antibodies might present lower sensitivity than those for phosphoserines (further discussed below).

Reviewer 2 raised the possibility that synaptic activity (a very dynamic process where NMDARs mainly containing GluN2A subunits participate) could be controlling PKD dephosphorylation through the activity of phosphatases. We also agree with his/her idea, based on the observation that the NMDARs inhibitor DL-AP5 was increasing PKD phosphorylation levels in the mature cultured neurons used (**see DL-AP5 panels in Figure 1c and 1d**). Inhibition of synaptic NMDARs by DL-AP5 in our neuronal cultures might therefore inactivate phosphatases and consequently increase PKD phosphorylation and activity. Text discussing these facts has been included under **Results (Page 8, line 193-197)**, see also related comments (**Page 9, line 218-223**).

Importantly, we were able to show in pY93-antibody immunoprecipitates that stimulation of neurons with excitotoxic concentrations of NMDA for 5 min increased the pY93 signal and that it disappeared after 1h (**novel panel Figure 1e**). This pattern of early stimulation of PKD phosphorylation followed by dephosphorylation at longer time points parallels the one observed for pS916 and pS744/748 (Figure 1a). Together, these new experiments presented in **Figure 1e and Supplementary Figure 2** show Src-induced phosphorylation of PKD Y93 by excitotoxic NMDAR activation. These data have been included under **Results (Page 8, line 187-197)**.

In sum, we have generated a novel phosphospecific mAb to detect PKD1 pY93, provide novel results on Y93 and Y463 phosphorylation in primary neurons in culture, and show its transient induction by NMDA-induced excitotoxicity.

COMMENT 2

Figure 3:

1) In Figure 3c and 3d, in order to exclude off-target effects, it is standard to either use as second shRNA, or to perform a rescue experiment (i.e. using a PKD1 from a different species). One of these needs to be included.

The point is well taken. As recommended, we have used a second shRNA for PKD1 (shPKD1a and shPKD1b) and obtained a similar result. **Figure 3c and 3d** and text in the manuscript have been changed accordingly under **Results (Page 12 line 284-286)**.

2) Figures 3f to 3i show only controls and should go to the Supplemental Data.

These panels have been moved from Figure 3 to **novel Supplementary Figure 6a-d**.

3) In Figure 3j, the tissues obtained with the PKD1 k/o or control mice should also be stained by

a - immunohistochemistry for pS910 and PKD1. These are important controls.

A **novel panel in Figure 3f** has been included showing immunohistochemistry for pS916. Given that immunoblot results indicate that the PKD antibody (Cell Signaling Technology #2052) we have used preferentially recognizes PKD1 in neuronal lysates (see new Supplementary Figure 1a), the immunohistochemistry images shown for this antibody in the previous version of the manuscript (**Figure 3f**) are indicative of PKD1 expression. Since pS916 antibody detects in immunoblot a doublet corresponding to PKD1 and PKD2 (new Supplementary Figure 1a), the immunohistochemistry signal that remains in the sections stained with pS916 from PKD1 KO brain might correspond to PKD2. Text in the manuscript has been changed accordingly under **Results (Page 12-13 line 296-301)**.

b - In addition, IHC for phosphorylation of oxidative stress-mediated tyrosine phosphorylations of PKD1 as well as markers for oxidative stress (i.e. 4-HNE) should be included.

We have tried different conditions to get IHC images with pY-antibodies (pY463 polyclonal antibody and pY93 mAb), in mouse brain, unfortunately without success. This could be due to their low efficiency recognizing these phosphosignals in endogenous PKD from neurons, as suggested by the fact that we only detected tyrosine phosphorylation of PKD1 in immunoblot analysis if immunoprecipitated extracts were used (see novel **Figure 1e** and **Supplementary Figure 2**).

From immunoblot analysis (discussed above in **COMMENT 1 – subpoint 5**), we know that detection of PKD tyrosine phosphorylation is difficult to assess in untreated neurons in culture. It could be possible that PKD phospho-specific tyrosine antibodies present lower sensitivity or affinity than the one detecting pSer916, or that only a subset of PKD molecules are tyrosine-phosphorylated. PKD participates in multiple functions, being ROS detoxification induced by Src-dependent mechanisms just only one of them. All these facts could contribute to hinder the observation of pY signal in brain sections. There is an additional comment we would like to point out here. This study defines this pathway as one utilized by neurons for ROS-detoxification under physiological conditions, where its activity is low but necessary to keep neurons healthy. Under pathological conditions (i.e. cerebral ischemia) levels of neuronal active PKD decrease. Therefore, we have not found a condition where this neuronal pathway can be potently induced by ROS overproduction. In conclusion, so far we do not have a model that produces increases in ROS levels that lead to PKD activation, where we could study the stimulation of this pathway in the brain in vivo. This reviewer mentioned an interesting report using a mouse model of pancreatic cancer induced by the potent oncogene K-Ras, that results in ROS production accompanied by significant increases in PKD p-Y95 (PMID 26947075), as a driver of pancreatic cancer development. In the mouse brain ischemic model we have used here, we observe neuronal ROS production (see next paragraph below) but PKD dephosphorylation. This PKD inactivation may be indeed contributing to the ROS increases registered during cerebral ischemia in neurons and to their oxidative stress damage and death. These notions are supported by additional novel experimental data showing that PKD KO brain presents higher oxidative stress damage in neurons (see below).

Regarding the brain tissue staining with markers for oxidative stress, reviewer 2 made the same suggestion including ischemic animals. Therefore, we performed immunofluorescence of brain sections from Sham and MCAO operated littermates of the two genotypes (PKD1^{flxed} and PKD1 KO). We used MDA, a lipid oxidation marker used to detect oxidative stress conditions (Anti-MDA JaICA; PMID 25786204). Importantly, **novel Figure 3i** shows that PKD1 KO sham animals contained significantly higher MDA signal than PKD1^{flxed} (**see images and quantitation analysis in the graph on the right**). MCAO provoked increases in this oxidative stress marker in both groups of mice, although PKD1 KO showed a 3-fold increase in this parameter compared to PKD1^{flxed} (**Figure 3i, images and quantitation**). Following reviewer's 2 suggestion, we also examined ROS production in PKD1^{flxed} and PKD1 KO cultured neurons after incubation with the oxidative stress-sensitive fluorescent dye DHE. We could determine that differences in ROS levels in those neurons did not reach significance under control conditions, but NMDA excitotoxic stimulation provoked significant increases in ROS production, being greater in PKD1 KO neurons (**novel Figure 3h and supplementary Figure 6h**). We can therefore conclude that elimination of PKD1 in neurons is accompanied by increases in ROS production. Information relative to ROS production in PKD1 KO neurons and brain has been added under **Results (Page 13 line 309-321)**.

COMMENT 3

Do antioxidants have the same effects that the PKD1 inhibitor in cells or in vivo?

As indicated above, we show now that PKD1 inactivation in neurons increases the presence of an oxidative stress marker (**novel Figure 3i**). These results therefore suggest strongly that antioxidants and PKD1 inhibition produce opposite effects.

COMMENT 4

In the introduction (page 4) the author's state: "Oxidative stress is an important activator of PKD1 in cellular models, but its capacity to activate this kinase in vivo has not been confirmed". This is actually not correct. PMID 26947075 have shown that the

ROS-PKD-NF-kappaB pathway is a driver of pancreatic cancer development. On page 5, a similar incorrect statement is made.

The point is well taken. These statements have been smoothed in the new version of the manuscript (**Page 4 line 94-95 and Page 5 line 113-114**), and the reference has been included under **Discussion (Page 20 line 472-474)**.

COMMENT 5

The quality of the immunofluorescence shown in Figure 2d needs to be improved.

This is a technically challenging task because availability and quality of sections obtained from human brain necropsies could sometimes be a handicap for obtaining good images. There are few samples and they are always obtained post-mortem (see post-mortem times in **Supplementary Table 1**). Quality relies heavily on post-mortem time and the temperature at which necropsies were kept before paraffin inclusion. We performed human brain sample immunofluorescence staining on 15 µm-thick paraffin sections mounted onto glass slides. We must point out that immunofluorescence and immunohistochemistry cerebral mouse images were obtained in free floating cryostat 25 µm-thick sections after brain fixation by intracardial PFA perfusion, postfixation and freezing. Therefore, we could not apply the same conditions to immunostain human and mouse brain. We are used to perform immunofluorescence and confocal microscopy analysis in our research and have tried several methods in order to get the best analysis from the human brain samples we had available. We provide **a new version of Figure 2d**, where we have tried to improve these images as much as possible, and are confident on the specificity of the signal obtained.

COMMENT 6

Molecular weight markers need to be added to the Western blots.

We thank the reviewer for this observation. Molecular weight markers have been added in all the Figures.

We hope that all the new data incorporated satisfactorily address the Reviewer's concerns.

NOTE: Main manuscript text, figure legends and Supplementary information text and legends contain information relative to new experiments, results and modifications labelled in “blue”.

Reviewer #2 (Remarks to the Author):

In this study García-Guerra et al investigate the role of PKD1 in excitotoxic neuronal injury and brain ischemia. The authors show that PKD1 is transiently activated after excitotoxic glutamate receptor activation in vitro and in two excitotoxic in vivo models, transient focal ischemia in mice and kainic acid neurotoxicity in rats. The authors hypothesize that post-activational down-regulation of PKD1 activity contributes to excitotoxic neuronal cell death. Using a comprehensive pathway analysis approach the authors show that loss of PKD1 activity possibly due to dephosphorylation by DUSP1 results in deactivation of the enzyme after excitotoxic NMDA challenge. PKD1 auto-phosphorylation was associated with increased NF-kappaB activity, reduced ROS production, and neuronal survival. Although the study uses a variety of assays to address the molecular components involved in PKD1-mediated cell survival, the assays are not applied consistently to all paradigms.

We thank the Reviewer for his/her careful reading of the manuscript and for raising important issues that we have taken into account to improve the revised version of our manuscript.

Specific concerns are:

COMMENT 1

1. The majority of studies, including those from the Schwaninger laboratory, show that excitotoxicity and ischemia induces neuronal NF-kappaB activation. These findings are in contrast to data presented in the current study. This leads to two essential questions:

QUESTION 1) Is the antibody to p65/RelA specific? This could be tested in RelA^{-/-} cells.

We are aware of the controversy regarding neuronal NF- κ B activation and therefore discuss it in the manuscript **introduction (page 5 lines 108-112)** and under **Discussion (Page 20 lines 492-498)**. To address this reviewer's concern regarding p65 antibody specificity, we cultured mouse embryo fibroblasts (MEFs) from control mice and *RelA*^{-/-} (p65 KO; obtained from Prof. Jianping Ye, Pennington Biomedical Research Center, Baton Rouge, LA, USA) on coverslips and used the p65 antibody (Santa Cruz Biotechnology # sc-372) used in our studies for immunofluorescence detection. MEFs were untreated or treated with TNF α to induce p65/RelA translocation to the nucleus. We have prepared a **new Figure for Reviewer 2**, showing that the antibody detects p65 in the cytosol of control untreated MEFs or in the nucleus of these cells after TNF α treatment. This antibody does not give any signal in p65 KO MEFs.

Figure for Reviewer #2. (a) Levels of NF- κ B RelA/p65 (p65) was assessed in total lysates (100 μ g) from wild type (WT) and *RelA*^{-/-} (p65 KO) mouse embryo fibroblasts (MEFs) by immunoblotting. Alpha-tubulin was used as loading control. **(b)** Detection of NF- κ B RelA/p65 (p65, green) localization by immunofluorescence in WT and p65 KO MEFs control or TNF α -treated.

Further supporting the specificity of the staining obtained with this p65 antibody, a similar pattern was observed with a phospho-p65 antibody (Cell Signaling Technology # 3031) (see **new Supplementary Figure 7b**, and information under **Results in Page 14 line 341-344**). Helpful information about all antibodies used in this study is now available in novel **Supplementary Table 2**.

QUESTION 2) Is nuclear translocation of RelA indeed reflective of NF- κ B activity. While nuclear translocation is clearly a prerequisite for NF- κ B mediated transcription, it is not always sufficient. Determining the S536 phosphorylation state might give further insight of NF- κ B transcriptional activity. In addition, EMSA assays of nuclear extracts might be useful to detect increased NF- κ B DNA binding activity.

The reviewer raises important issues. To address them, we carried out immunofluorescence detection of phosphorylated pS536-p65/RelA (p-p65) and quantitation analysis of its nuclear localization in untreated or NMDA treated neurons. As shown in novel Supplementary Figure 7b, p-p65 is constitutively localized in neuronal nuclei and its nuclear presence decreases significantly after 2h of NMDARs overstimulation. Information has been added under **Results (Page 14 line 341-344)**. As an approach to measure/examine NF- κ B transcriptional activity, we performed luciferase assays using a luciferase reporter with three NF- κ B response elements cloned before the TK promoter driving luciferase gene transcription. We observed that luciferase activity significantly decreased in NMDA-treated neurons (novel panel Figure 4c). Text including these results has been added under **Results (Page 14 line 345-347)**. Together, these new data indicate that neurons present basal NF- κ B activity that is downregulated under excitotoxic conditions.

COMMENT 2

2. The mechanism of NF-kappaB induction by PKD1 remains elusive. What are the molecular targets of PKD1? Is it acting directly on IKK or other upstream kinases involved in NF-kappaB activation?

PKD1 forms a complex with IKK2 and controls IKK activity (PMID:12505989). In addition, PKD1 siRNA decreases IKK phosphorylation at Ser180/Ser181 in the activation loop of its catalytic domain, thus diminishing IKK activity (PMID:16166634). These data correlate well with our results showing that the expression of constitutively active PKD1 in neurons enhances IKK phosphorylation at those activatory residues (Figure 5b). This information has been included (**Page 5 line 104-108**) and these two references are cited therein.

COMMENT 3

3. ROS production should also be tested in PKD1 KO neurons. In addition, *in vivo* data are needed to determine whether the regulation of ROS generation found in cultured neurons are also observed in **the ischemic and kainite model *in vivo***. In addition, the 24 hours endpoint seems rather late to assess pathways involved in cell death in the ischemic model because excitotoxic neuronal loss might already been maximal at this time.

These are important issues and we have made a major effort to address them:

- We have now examined ROS production in PKD1^{flxed} and PKD1 KO cultured neurons (obtained from littermates) after incubation with the oxidative stress-sensitive fluorescent dye DHE. We found that ROS levels under control conditions did not reach significant differences in neurons from the two genotypes. However, NMDA excitotoxic stimulation provoked significant increases in ROS production, being greater in PKD1 KO than in control neurons (**novel Figure 3h and supplementary Figure 6h**). Information has been added under **Results (Page 13 line 309-315)**. We therefore conclude that elimination of PKD1 in cultured neurons is accompanied by increases in ROS production under excitotoxic conditions.

- Regarding ROS production *in vivo*, we performed immunofluorescence of brain sections from Sham and MCAO operated littermates of PKD1^{flxed} and PKD1 KO mice. We used MDA, a lipid oxidation marker sensitive to oxidative stress (Anti-MDA JaICA; PMID 25786204). Importantly, **novel Figure 3i** shows that PKD1 KO sham mice contained significantly higher MDA signal than PKD1^{flxed} mice (**see images and quantitation analysis in the graph on the right**). Although MCAO increased MDA staining in both groups of mice, PKD1 KO showed a 3-fold increase in this parameter compared to PKD1^{flxed} animals (**Figure 3i, images and quantitation**). Information has been added under **Results (Page 13 line 315-321)**. Unfortunately, we could not detect MDA immunostaining in CA1 regions of rat brain injected with GFP or with active PKD1 after kainic acid administration.

- As suggested by this reviewer and reviewer #3, we have also performed the ischemic MCAO model for shorter times after the ischemic insult (1 hour occlusion and 5 hours reperfusion). Of note, we obtained similar results regarding neuronal PKD inactivation and loss of NF- κ B nuclear signal in the penumbra area (novel Supplementary Figure 4). Information has been added under **Results (Page 11 line 261-262)**. The ischemic core in the striatal area was already severely affected, even though reperfusion time was shortened from 24h to 5h, and no signal was detected for active PKD1 and NF- κ B (not shown).

COMMENT 4

4. Fig 1d: It is not clear why AP5 treatment increases PKD1 activity at baseline. Does synaptic activity inhibit PKD1?

This is an interesting observation. A recent report describes that PKD activity regulates synaptic plasticity in dendritic spines by regulating actin dynamics (PMID:26304723). To the best of our knowledge, whether synaptic activity can inhibit PKD1 remains unknown. However, it has been reported that stimulation of synaptic NMDARs activates the phosphatase PP1 (PMID:24189275). Since we show that PP1 is involved in PKD1 dephosphorylation (Figure 1g), inhibition of synaptic NMDARs by AP5 in our neuronal cultures might inactivate PP1 and consequently increase PKD phosphorylation and activity. We mention this possibility in the revised manuscript under **Results (Page 7 line 160-163)**.

COMMENT 5

5. Activity and activation are used synonymously. Kinase phosphorylation at one residue is not necessarily predictive of its activity.

We understand the Reviewer's concern about the use of activity and activation. We have been working with PKD since 1995 and have been involved in the identification and characterization of several activation mechanisms. One of them consists in the phosphorylation of PKD1 catalytic activation loop by protein kinase C in two residues (serines 744 and 748; Iglesias et al JBC PMID:9765302). This phosphorylation is activatory. Although pS744/748 leads to PKD1 activation, it does not predict PKD1

activity, as pointed by the reviewer. It should be noted however that this phosphorylation normally parallels activation of the kinase. By contrast, Ser916 is an autophosphorylation site (PMID:10473617), and therefore p-S916 signal is usually taken by researchers in the PKD field as a readout of PKD1 kinase activity. We show in **Figure 1a** that neurons present basal phosphorylation at S744/748 and S916, parameters that indicate activation by PKC-transphosphorylation and activity by PKD-autophosphorylation, respectively. We also show that excitotoxicity decreases phosphorylation at both S744/748 and S916. In sum, we believe we can talk about PKD activation when we see increases in PKD1 activity determined by its final step of S916 autophosphorylation.

COMMENT 6

6. The activity of the constitutive active PKD1 mutants should be tested in an in vitro kinase assay.

As discussed above, pS916 levels can be taken as readout of PKD1 activity. Since the pS916 signal shown in **Figure 5b** is extremely high in GFP-PKD1-Ca transduced neurons, it is indicative of high PKD1 activity. Nevertheless, following this reviewer's suggestion, we performed an in vitro kinase assay that shows the constitutive activity of this mutant compared to PKD1 WT or an inactive kinase dead mutant (PKD1 KD) (see **novel Supplementary Figure 8a**). This information has been added under **Results (Page 15 line 371-372)**.

COMMENT 7

7. Cellular experiments using phosphatase inhibitors are more complex than stated by the authors. Increased phosphorylation of PKD1 upon inhibitor treatment might not indicate that these changes are due to direct activity of the respective phosphatases on the target. Most likely, some of these inhibitors are also resulting in increased activity of upstream protein kinases which could potentially result in increased PKD1 phosphorylation.

We agree with the reviewer that different compounds, including phosphatase inhibitors, could produce unwanted off-target effects difficult to explore experimentally, and that

inhibition of phosphatases will result in increases of phosphorylation by kinases acting on the same substrate. However, these inhibitors are widely used to investigate how signalling pathways work. Here we have used them as tools to get insight into PKD inactivation mechanisms that were so far unknown. That is why we have combined when possible the use of inhibitors together with sh-RNA approaches to further support or rule out the participation of a certain phosphatase on PKD phosphorylation (i.e. DUSP1 and STEP, respectively).

COMMENT 8

8. In some vitro experiments the dosage of NMDA is not evident (e.g figure 1b, 5c).

We thank the reviewer for this observation. Most of our studies are performed stimulating with NMDA (50 μ M) and glycine (10 μ M), a treatment we refer as “NMDA” for simplifying purposes. We have introduced this information under Results (**Page 6 lines 133-134**) and also in legend of Figure 1a. Dose responses used (Supplementary Figure 1d) are indicated therein.

COMMENT 9

9. IKK1 and IKK2 might have opposing effects on NF-kappaB activity. It is not clear which of the two enzymes has been investigated in this study. Furthermore, the NF-kappaB activation should be tested on the transcriptional level by monitoring the mRNA expression levels of selected NF-kappaB-dependent genes including SOD2 in vitro and in vivo.

We have used antibodies that detect phosphorylated IKK1 and IKK2 (Biorbyt # orb127876), IKK2 (Cell Signaling Technology # 2678) as well as a general IKK inhibitor. We cannot therefore exclude one or the other with the studies we have performed. Following the reviewer’s recommendations, we have determined mRNA levels of *Sod2* and those of *Bdnf*, a prosurvival neurotrophin whose transcription can be regulated by NF- κ B (PMID:18040799). Using RT-qPCR we have found that the mRNA levels of both genes are downregulated by excitotoxicity in cultured neurons (**Novel panel Figure 4d**) (see text under **Results Page 14-15 line 347-351**). Considering that our study is based on the identification of PKD/NF- κ B signalling only in neurons, we

did not try to carry out RT-qPCR studies after extracting RNA from brain preparations/homogenates. Results obtained from brain homogenates could lead to misleading conclusions due to the input from multiple brain cell populations as we discussed in the initial version of this manuscript (**Discussion, page 20 lines 494 – 498**). This is the reason why this set of new data comes only from in vitro models.

COMMENT 10

10. Were PKD1flox/flox and PKD1 KO mice derived from littermates? What is the genetic background of these animals.

Mice are C57BL/6 and PKD1^{flox^{ed}} and PKD1 KO mice used for comparison analysis through the manuscript were littermates. A sentence related to this issue has been added to the **Methods** section (**Page 23-24, lines 556-560**).

- The authors should also provide an assessment of macroscopic brain vascular anatomy of PKD1 KO mice.

We thank the reviewer for this suggestion. We did an intracardial perfusion of Evans blue to macroscopically label brain vascular anatomy in control PKD1^{flox^{ed}} and PKD1 KO mice littermates and found no significant differences between genotypes after *ex vivo* whole brain analysis (**novel Supplementary Figure 6g**). **A sentence has been included under Results (Page 13 line 303-304).**

COMMENT 11

11. The assessment of histological sections should have been performed in a stereological fashion. It is not clear how representative the current analysis is.

This is an important observation and we must apologize if we did not explain properly how stereological analysis of histological sections was performed. Text under **Methods, Immunohistochemistry** section has been added to clarify this important point (**Page 34, lines 813-824, and page 35 lines 832-849**).

We hope that all the new data incorporated satisfactorily address the Reviewer's concerns.

NOTE: Main manuscript text, figure legends and Supplementary information text and legends contain information relative to new experiments, results and modifications labelled in “blue”.

Reviewer #3 (Remarks to the Author):

The authors' findings are interesting and novel. However, there are several concerns that reduce enthusiasm for the manuscript.

We thank the Reviewer for his/her comments on the interest and novelty of our findings. His/her criticisms and suggestions have been very valuable and have helped us to improve the manuscript significantly. We have performed the experiments requested to further support our conclusions.

COMMENT 1

1. The authors are advised not to use the term “excitotoxicity-induced oxidative stress” as it implies that oxidative stress is the key mediator of excitotoxic neuronal death. Excitotoxicity can manifest as necrosis which results from rapid massive Na⁺ influx and cell swelling and membrane rupture, or as apoptosis triggered by Ca²⁺ influx and uptake into mitochondria / PTP opening / cytochrome c release / caspase 3 activation... While oxidative stress certainly occurs in excitotoxic neuronal death, it is not a pivotal early event.

Following this reviewer's recommendation, we have eliminated this term (see Abstract **Page 3 lines 60-61**).

- Also, while the authors focus on genes encoding antioxidant enzymes as NF-κB targets in neurons (which is true), NF-κB also induces genes encoding anti-apoptotic Bcl-2 family members and Ca²⁺-regulating proteins that can protect against excitotoxicity.

This is an interesting point and we therefore determined mRNA levels of the anti-apoptotic Bcl2 family members *Bcl-2* and *Bcl-xL*. In our analysis we also included the

mRNA quantification of *Sod2* and *Bdnf*, a prosurvival neurotrophin whose transcription can be regulated by NF- κ B (PMID:18040799). Using RT-qPCR, we did not observe significant changes on transcripts for *Bcl-2* and *Bcl-xL* induced by excitotoxicity in cultured neurons (**Figure for Reviewer 3**).

Figure for Reviewer #3. *Bcl2* and *Bcl-xL* mRNA levels measured by RT-qPCR in primary cortical neurons untreated or stimulated with NMDA for 2h. Data are shown as mean \pm s.e.m (n=3 independent experiments).

In contrast, *Sod2* and *Bdnf* levels were significantly downregulated after NMDA treatment (**Novel panel Figure 4d**); (**Page 14-15 line 347-351**). These data indicate that not all NF κ B target genes are subjected to the same mechanisms of regulation. In addition, given that this model of excitotoxicity induces necrotic death, it is conceivable that it does not affect anti-apoptotic molecules.

COMMENT 2

2. PP1 is activated by Ca²⁺ and it is likely Ca²⁺ influx that causes dephosphorylation of PKD1.

The reviewer raises an important issue. Although one of the major effects of NMDAR overstimulation is massive Ca²⁺ influx into the neurons, we did not explore whether PKD1 dephosphorylation depends on the entry of this ion. We have treated now cultured neurons with the calcium ionophore A23187 for different periods of time. We found a very similar pattern of PKD phosphorylation/dephosphorylation to the one produced by NMDA, registering increases after 5 min of ionophore addition and decreases at longer time points (**Novel Supplementary Figure 1i**). In a complementary approach, we did the opposite, and blocked Ca²⁺ entry by pre-treating the cultures with

the chelator EGTA before overstimulating NMDARs. The result showed that EGTA not only clearly blocked NMDA-induced PKD dephosphorylation, but it also increased basal PKD phosphorylation (**Novel Supplementary Figure 1h**). As commented to reviewer 2, it has been reported that physiological stimulation of synaptic NMDARs activates PP1 phosphatase (PMID:24189275). NMDARs antagonists such as DL-AP5 or calcium chelators such as EGTA would also block Ca²⁺ entry through the synaptic NMDARs that will be functioning in a mature neuronal culture, and consequently PP1 activity. According to this and to our results on PP1 being a PKD phosphatase, we observe that both EGTA and DL-AP5 increase PKD1 phosphorylation (**see DL-AP5 panels in Figure 1c and 1d, and Supplementary Figure 1h**). Together these data indicate, as anticipated/predicted by the reviewer, that Ca²⁺ influx causes dephosphorylation of PKD1. Information relative to this section has been added under **Results (Page 7 line 160-171)**.

COMMENT 3

3. Figure 2a and b. Most of the striatal neurons in the region examined are killed by the ischemic insult. It is therefore not surprising that little or no p-PKD is seen. Animals should be killed at earlier time points 2 – 8 hours after the ischemic insult, before the neurons are dead.

As suggested by this reviewer and by reviewer #2, we have also performed the ischemic MCAO model for shorter times after the ischemic insult (1 hour occlusion and 5 hours reperfusion). Of note, we obtained similar results regarding neuronal PKD inactivation and loss of NF- κ B nuclear signal in the penumbra area (**Novel Supplementary Figure 4**). Information has been added under **Results (Page 11 line 261-262)**. The ischemic core in the striatal area was already severely affected, even though reperfusion time was shortened from 24h to 5h, and no signal was detected for active PKD1 and NF- κ B (not shown).

We believe that observing the same mechanisms after short and long reperfusion is not surprising because it is well established that excitotoxic cell death mechanisms are triggered in the penumbra area as a result of the massive release of glutamate from dying or dead neurons in the neighbouring ischemic core. That is why it is quite reasonable to examine the penumbra area to study the effects of excitotoxicity. Indeed,

we observe very similar results in the penumbra area to those found in the *in vitro* excitotoxicity model. In contrast, neuronal death and damage in the ischemic core is very fast and severe, and it is primarily caused by energy failure due to the lack of oxygen and glucose (and not to toxicity induced by high glutamate). It was therefore important in our initial study to analyse PKD1 and NF- κ B not only in the penumbra area, but also in the ischemic core because we did not know the effect that oxygen and glucose deprivation could have in this signalling pathway in the ischemic brain.

COMMENT 4

4. It is clear from the images of cultured neurons shown in Figures 3 and 4, and from an extensive literature of published studies of excitotoxicity in primary neuronal cultures, that with the concentration of NMDA used the neurons are dying of rapid (within 1 – 2 hours) necrosis (cell swelling and rapid neurite beading and fragmentation). Excitotoxic necrosis cannot be prevented with caspase inhibitors, and I expect that caspase inhibitors will not protect the neurons in the present study. When neurons die by excitotoxic apoptosis, the cell death is delayed for 12 -24 hours, and the cells shrink in size (this death can be inhibited with caspase inhibitors).

The reviewer is right. In fact, while investigating the *in vitro* excitotoxicity model used in this study, we found that many of the processes linked to neuronal death are not dependent on caspase activation but on calcium entry and calpain activation (i.e PMID:19759287). In any case, we performed additional analysis of neuronal viability in the presence of the caspase inhibitor zVAD-fmk and found that it did not improve neuronal survival upon NMDA treatment (**Novel Supplementary Figure 1f**). This information has been added under **Results (Page 7 lines 151-152)**.

COMMENT 5

5. When neurons die by excitotoxic necrosis their nuclei often exhibit non-specific immunoreactivity with many different antibodies. This may be the case in the present study, resulting in nuclear staining with the NF- κ B subunit antibody.

We understand the reviewer's concern. However, NF- κ B staining localizes at the nuclei of healthy neurons in this study and excitotoxicity produces a significant decrease of

this nuclear staining. Therefore, we do not observe increase in nuclear staining under excitotoxic conditions. A New Supplementary Table 2 with the information relative to the antibodies used in our study has been included.

COMMENT 6

6. The excitoprotective effect of the dephosphorylation-resistant PKD1 mutant is impressive. This may be due to an effect of p-PKD1 on Ca²⁺ influx or buffering. Ca²⁺ imaging studies are required to rule this in or out.

The reviewer raises a very interesting issue. To address it, we have incubated GFP or GFP-PKD1 Ca transduced neurons with cell permeant Fura Red, AM, and performed Ca²⁺ imaging analysis by *in vivo* confocal microscopy before and after addition of excitotoxic concentrations of NMDA. Results show no significant differences in the Ca²⁺ influx peak between GFP or GFP-PKD1 Ca transduced neurons (Novel Supplementary Figure 8b). This information has been added under **Results (Page 16 line 389-391)**.

We hope that all the new data incorporated satisfactorily address the Reviewer's concerns.

REVIEWERS' COMMENTS:

Reviewer #1 (Remarks to the Author):

The authors answered all my points. This is an excellent paper and I would recommend it for publication.

Reviewer #2 (Remarks to the Author):

In this revised submission the authors have addressed several concerns raised by this reviewer. Importantly they have demonstrated the specificity of reagents, included new time points of analysis in the stroke model, and assessed ROS production in vitro and in vivo. In addition, the authors have clarified inconsistencies regarding the experimental protocols and animal usage. In summary, the manuscript has been greatly improved and the current study clearly defines a role for PKD1 in excitotoxic neuronal injury.

Reviewer #3 (Remarks to the Author):

The authors have addressed my concerns satisfactorily